# ROBOTWIN 2.0: A SCALABLE DATA GENERATOR AND BENCHMARK WITH STRONG DOMAIN RANDOMIZATION FOR ROBUST BIMANUAL ROBOTIC MANIPULATION

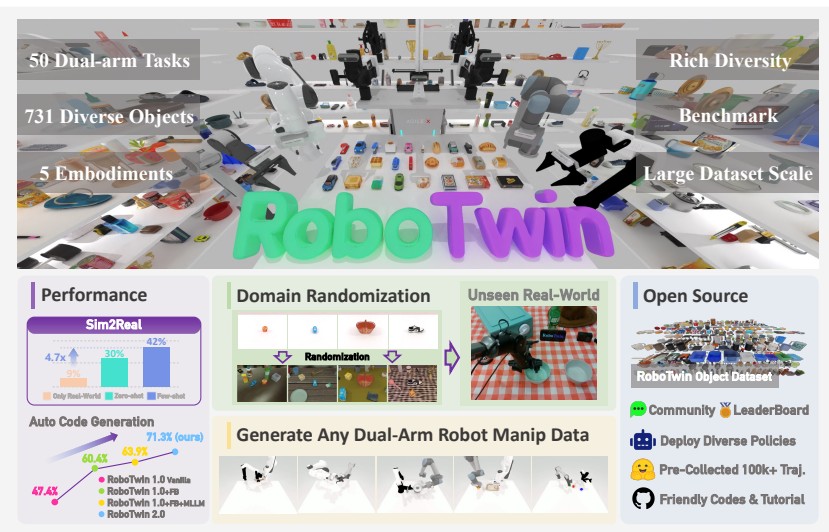

Figure 1: **RoboTwin** 2.0 uses an MLLM-driven pipeline for automatic data synthesis and domain randomization to boost policy performance, and provides a 50-task bimanual benchmark with RoboTwin Object Dataset.

## ABSTRACT

Synthetic data generation via simulation represents a promising approach for enhancing robotic manipulation. However, current synthetic datasets remain insufficient for robust bimanual control due to limited scalability in novel task generation and oversimplified simulations that inadequately capture real-world complexity. We present **RoboTwin** 2.0, a scalable framework for automated diverse synthetic data generation and unified evaluation for bimanual manipulation. We construct RoboTwin-OD, an object library of 731 instances across 147 categories with semantic and manipulation labels. Building on this, we design a expert data generation pipeline by utilizing multimodal large language models to systhesize task-execution code with simulation-in-the-loop refinement. To improve sim-to-real transfer, RoboTwin 2.0 applies structured domain randomization over five factors (clutter, lighting, background, tabletop height, language instructions). Using this approach, we instantiate 50 bimanual tasks across five robot embodiments. Experimental results demonstrate a 10.9% improvement in code-generation success rates. For downstream learning, vision-language-action models trained with our synthetic data achieve 367% performance improvements in the few-shot setting and 228% improvements in the zero-shot setting, relative to a 10-demo real-only baseline. We further evaluate multiple policies across 50 tasks with two difficulty settings, establishing a comprehensive benchmark to study policy performance. We release the generator, datasets, and code to support scalable research in robust bimanual manipulation.

# 1 INTRODUCTION

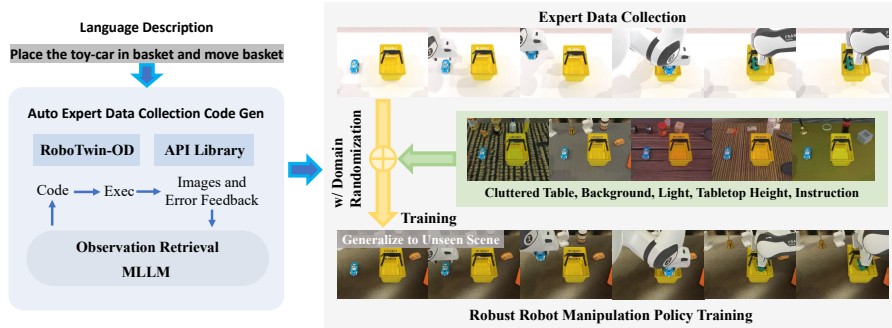

Figure 2: **Our Pipeline.** Built on RoboTwin-OD and skill APIs, an MLLM guides code generation with simulation feedback to produce expert programs and domain-randomized trajectories.

Bimanual robotic manipulation is essential for complex tasks such as collaborative assembly, tool use, and handovers. Training generalizable bimanual policies, particularly vision-language-action (VLA) foundation models Long et al. (2025), requires datasets that are high quality, diverse, and large scale. Without sufficient variation in object geometry, scene clutter, lighting, instruction language, and robot embodiments, learned policies overfit and generalization degrades across environments and hardware. However, collecting real-world demonstrations at scale remains costly, time-intensive, and logistically difficult, especially when targeting broad task, object, and embodiment coverage.

Simulation has become an effective way to scale multimodal data collection and enable sim-to-real transfer Mu et al. (2025); Deng et al. (2025). However, prevailing pipelines exhibit three persistent limitations: (i) the absence of automated quality control, which admits execution failures and weak grasps that degrade learning; (ii) shallow domain randomization, producing overly clean, homogeneous scenes that neglect clutter, illumination changes, and instruction ambiguity—factors critical for robust transfer; and (iii) limited cross-embodiment coverage, despite substantial differences in kinematics and grasp strategies across bimanual platforms. For example, low-DoF systems such as Piper tend to favor lateral grasps, whereas high-DoF arms like Franka support top-down precision grasps. Current synthetic datasets rarely encode these embodiment-specific affordances and task constraints, limiting generality.

To address these challenges, we introduce **RoboTwin 2.0**, a scalable simulation-based framework for generating high-quality, diverse, and realistic datasets for bimanual manipulation. The framework comprises: (1) an automated expert pipeline that uses multimodal large language models (MLLMs) with simulation-in-the-loop feedback to validate and refine task execution code; (2) comprehensive domain randomization over language, clutter, background textures, lighting, and tabletop layouts to improve sim-to-real transfer and policy generalization; and (3) embodiment-aware adaptation that annotates object affordances and generates robot-specific action candidates for heterogeneous dual-arm kinematics. Building on these components, we introduce three new resources to support scalable research in bimanual manipulation: (1) the RoboTwin-OD asset library, comprising 731 annotated object instances across 147 categories; (2) an automated data generation pipeline with comprehensive domain randomization and a pre-collected, open-source dataset of expert trajectories spanning 50 tasks across five dual-arm robot platforms; and (3) a benchmark for evaluating policy generalization to cluttered environments and open-ended language goals. Together, these resources enable the community to train and evaluate robust bimanual manipulation policies under conditions that closely reflect real-world complexity and diversity.

In summary, our main contributions are as follows: (1) We develop an automated expert data generation framework that integrates MLLMs with simulation-in-the-loop feedback to ensure high-quality, expert-level trajectories; (2) We propose a systematic domain randomization strategy that enhances policy robustness by increasing data diversity and sim-to-real generalization; (3) We introduce an embodiment-aware adaptation mechanism that generates robot-specific manipulation candidates based on object affordances; (4) We release the RoboTwin-OD, a large-scale pre-collected multi-embodiment domain-randomized trajectory dataset, a scalable bimanual data generator, and a standardized evaluation benchmark to support scalable training and evaluation of generalizable policies across different robot embodiments, scene configurations, and language instructions.

## 2 METHOD

Figure 2 overviews the RoboTwin 2.0 pipeline. A task–code generation module employs MLLMs with simulation-in-the-loop feedback to synthesize executable plans from natural-language instructions. The module is grounded in a large object asset library (RoboTwin-OD) and a predefined skill library, enabling scalable instantiation across diverse objects and manipulation scenarios. A comprehensive domain-randomization scheme along language, visual, and spatial dimensions further expands coverage, producing diverse, realistic demonstrations and policies robust to real-world variability.

### 2.1 EXPERT CODE GENERATION VIA MLLMs AND SIMULATION-IN-THE-LOOP FEEDBACK

We adopt a closed-loop architecture that couples code generation with multimodal execution feedback (Fig. 3), in contrast to pipelines that depend on manual priors or omit feedback Hua et al.; Wang et al. (2023). The system comprises two agents: a code-generation agent that translates natural language instructions into executable programs, and a vision–language model observer that monitors execution in simulation, detects failures and suggests corrections. Iterative integration of these signals proceeds until a predefined success criterion is met or a budget limit is reached, yielding robust, self-improving expert trajectories with minimal human supervision and enabling zero-shot dual-arm manipulation beyond primitive pick and place.

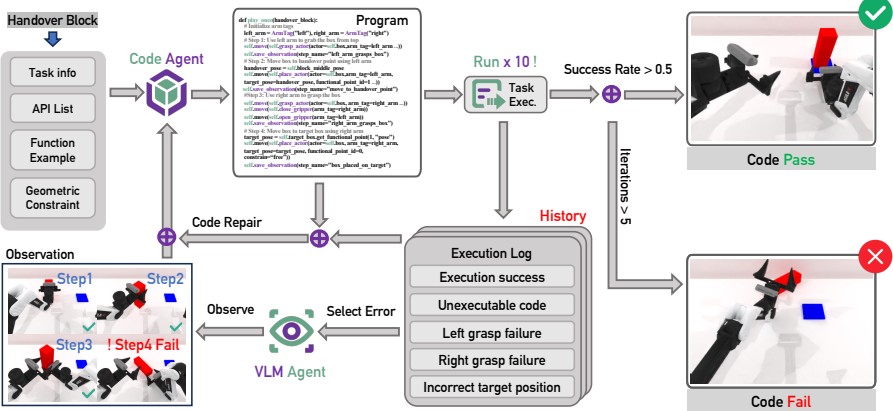

Figure 3: **Expert Code Generation Pipeline.**

**Input Specification.** The code-generation agent is conditioned on four inputs: (1) a general API list; (2) example function calls; (3) a hierarchical constraint specification; and (4) task information. Each task is defined by a name (e.g., *Handover Block*) and a natural-language objective description. These components jointly guide the synthesis of Python code for task execution.

**Initial Code Generation.** The code-generation agent synthesizes an initial Python program conditioned on the provided task inputs. It models the program synthesis process as a structured prediction problem over the space of available API calls, leveraging natural language understanding and few-shot prompting from task-specific examples. The generated code specifies a stepwise sequence of robot actions designed to accomplish the target manipulation objective.

**Simulated Execution and Logging.** Each iteration executes the program ten times in simulation to account for stochasticity in dynamics, control, and scene layout. After each batch, the system produces a structured log that records trial outcomes and labels failure cases by cause, such as unexecutable code, left/right grasp failure, or incorrect object placement.

**Multimodal Observation and Error Localization.** During execution, a vision–language model (VLM) monitors all ten trials and performs per-frame analysis to assess stepwise success and localize failures. Beyond temporal localization, the VLM attributes failure modes to flawed logic, incorrect API usage, or other systemic causes. This diagnosis enables repairs that target root causes rather than surface symptoms. Details are provided in Appendix A.10.4.

**Code Repair and Iterative Refinement.** The agent integrates execution logs and VLM diagnostics to edit failure-prone instructions, re-testing the program each iteration. The process stops upon meeting

a success-rate threshold over ten runs in one iteration, or after five consecutive failures, producing expert-level code with minimal supervision and avoiding indefinite refinement.

## 2.2 DOMAIN RANDOMIZATION FOR ROBUST ROBOTIC MANIPULATION

To enhance robustness to real-world variability, we randomize five dimensions: (1) cluttered distractors, (2) background textures, (3) lighting, (4) tabletop height, and (5) language instructions. This systematic augmentation broadens the training distribution and, critically, equips manipulation policies with stronger generalization to unseen scenes and instructions (Fig. 4a).

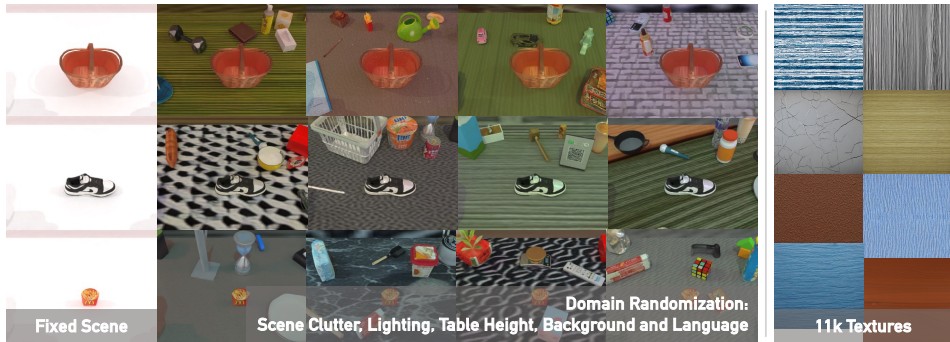

(a) Visualization of Domain Randomization    (b) Texture Library

Figure 4: **Visualization of domain randomization and our texture library.**

**Scene Clutter.** To improve robustness to environmental variation, we augment tabletop scenes with task-irrelevant distractors sampled from RoboTwin-OD (Section 3.1). Object-level placement annotations enable a generic API for semantically valid insertion. Physical plausibility is enforced through collision-aware placement and precomputed volumes. To prevent spurious ambiguity, distractors that are visually or semantically similar to task-relevant objects are excluded during sampling. This procedure yields diverse yet unambiguous cluttered scenes for training.

**Diverse Background Textures.** We randomize tabletops and backgrounds using a curated texture library. We first collect 1,000 surface descriptions via LLM prompting and web search, then generate 20 images per description with Stable Diffusion v2 Rombach et al. (2022) (20,000 images in total). Human-in-the-loop filtering yields 11,000 high-quality textures. This library is used in simulation to increase visual diversity and mitigate overfitting to clean synthetic scenes (Fig. 4b). We further show in Appendix A.18 that the distribution of our texture library closely matches that of real-world textures.

**Lighting Variation.** Real scenes vary in color temperature, source type, count, and placement, altering appearance and reflections and challenging vision-based manipulation. We randomize light color, type, intensity, and position within physically plausible ranges. As shown in Fig. 4a (second row), changes in color temperature markedly affect appearance (e.g., warm vs. cool light on a shoe). Training under these variations improves robustness to real-world illumination shifts.

**Tabletop Heights.** We uniformly randomize table height within a plausible range in simulation, strengthening the policy's robustness to variations in table height.

**Trajectory-Level Diverse Language Instructions.** We employ a MLLM to generate task templates and multiple object descriptions that capture geometry, appearance, and part-level attributes. Each task and object has several alternative phrasings that can be combined; for each trajectory, we sample from these pools to compose the instruction. For *Move Can Pot*, the template "Use a to place A to the left of B" may yield "Use left arm to place sauce can to the left of gray kitchenpot" or "Use left arm to place white plastic lid sauce can to the left of kitchenpot for boiling and cooking." This combinatorial augmentation produces a large, linguistically varied instruction set and improves generalization to unseen language and scene configurations (Appendix A.11, A.12).

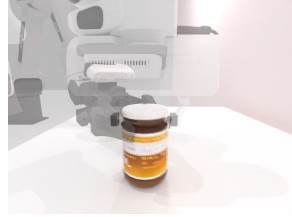 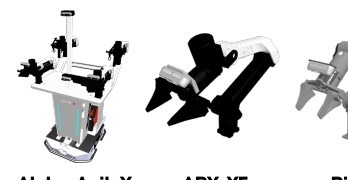 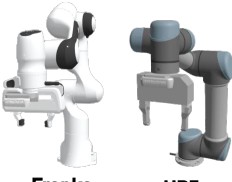

Figure 5: **Diverse Behaviors**.      Figure 6: **Five RoboTwin 2.0 Embodiments**.

## 2.3 EMBODIMENT-AWARE GRASP ADAPTATION

Differences in DoF and kinematics result in different reachable workspaces and preferred strategies for a given task. In grasping a can, Franka often adopts an overhead approach, whereas the lower-DoF Piper favors lateral grasps; consequently, required approaches vary across embodiments (Fig. 5). To model this variation, we annotate each object with candidate manipulation poses that span multiple grasp axes and approach directions, capturing both manipulation diversity and robot-specific preferences. We further improve feasibility via angular perturbations oriented to high-reachability directions. For each object, candidate grasps are generated from preferred operation directions, randomized pose perturbations, and parallel motion-planning attempts. In experiment A.2, embodiment-aware augmentation raises automated data-collection success by $8.3\%$ on average, with gains concentrated on low-DoF platforms (Aloha-AgileX +13.7%, Piper +22.7%, ARX-X5 +5.6%), while high-DoF arms (Franka, UR5) exhibit minimal change, consistent with greater kinematic flexibility.

## 3 ROBOTWIN 2.0 DATA GENERATOR, BENCHMARK AND RDDATASET

### 3.1 ROBOTWIN-OD: ROBOTWIN OBJECT DATASET

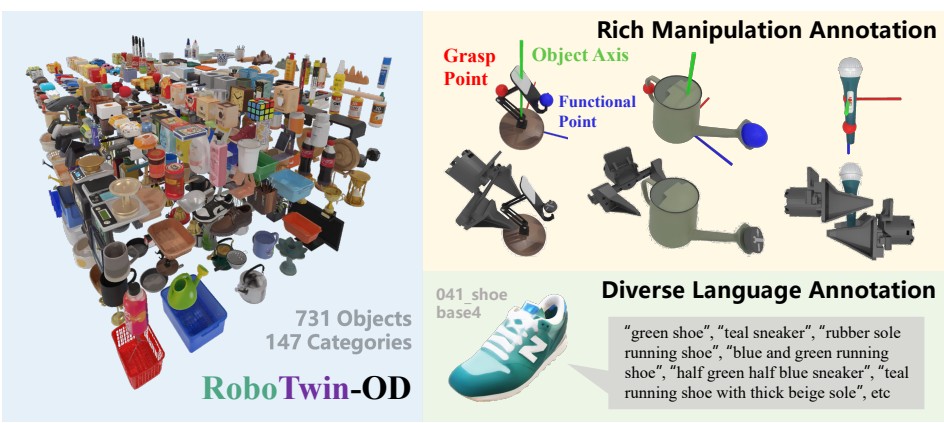

Figure 7: **RoboTwin-OD.** A large-scale object dataset with rich annotations.

We build RoboTwin-OD, an object dataset with rich semantics covering 147 categories and 731 objects: 534 in-house instances across 111 categories reconstructed from RGB-to-3D via the Rodin platform rod, followed by convex decomposition and mesh merging for physically accurate collisions; 153 objects from 27 categories in Objaverse Deitke et al. (2023); and 44 articulated instances from 9 categories in SAPIEN PartNet-Mobility Xiang et al. (2020). All sources support cluttered scenes, with Objaverse enhancing the visual and semantic diversity of distractors. We also curate a texture library for surfaces and backgrounds using generative models with human-in-the-loop filtering. To support language grounding and robustness across diverse objects, we deploy an automated description generator with human verification, producing 15 annotations per object that vary in shape, texture, function, part structure, and granularity. For object-centric interaction, we annotate key point–axis information, including placement points, functional points, grasp points, and grasp axes, to encode affordances. Combined with our manipulation API library, these annotations enable generalizable grasp execution in simulation.

## 3.2 50 Tasks for Data Generation and Benchmarking

Building on automated task generation, embodiment-adaptive synthesis, and the RoboTwin-OD asset library, we define 50 dual-arm collaborative manipulation tasks. Data collection and evaluation are supported on five robot embodiments, enabling comprehensive cross-embodiment benchmarking; representative keyframes are shown in Fig. 8. We also release a pre-collected corpus of 100,000+ dual-arm trajectories across these tasks in RoboTwin 2.0.

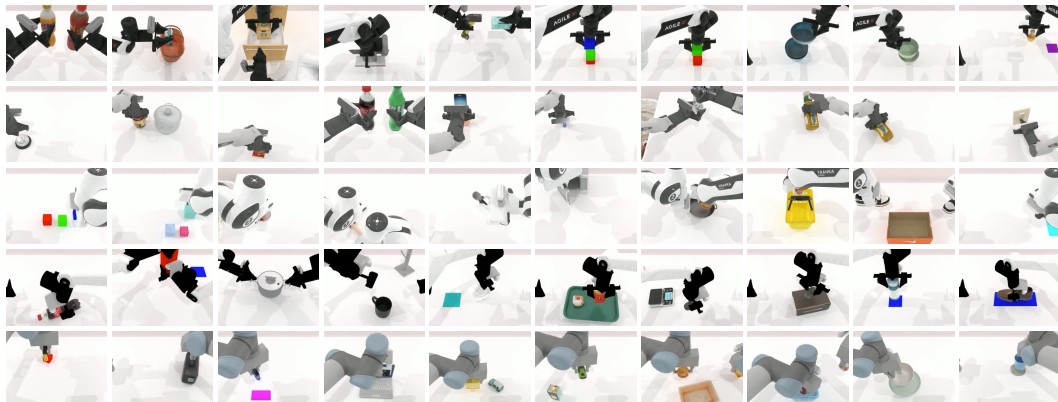

Figure 8: **50 Bimanual Manipulation Tasks with Multi-Embodiment Support per Task**.

## 4 EXPERIMENT

We design experiments to evaluate RoboTwin 2.0 along four dimensions: (1) automating the generation of high-quality expert code for novel manipulation tasks; (2) establishing RoboTwin 2.0 as a standardized benchmark for policy generalization across tasks, scenes, and embodiments; (3) improving policy robustness to environmental variation via diversified training data; and (4) demonstrating sim-to-real transfer, whereby RoboTwin 2.0 enables deployment on real robots and confers strong policy generalization to variations in scene composition and appearance.

### 4.1 EVALUATION OF AUTOMATED EXPERT CODE GENERATION

To assess whether closed-loop generation improves the quality and efficiency of expert programs, we evaluate the system on 10 manipulation tasks, each specified by a natural-language instruction. For each configuration, the code-generation agent emits multiple candidate programs that are executed in simulation to capture stochasticity in dynamics, control, and perception; task success is defined as the mean success rate over all executions. Performance is measured by **ASR** (average success rate), **Top5-ASR** (average of top-5 success rate), **CR-Iter** (average refinement iterations), and **Token** (average tokens in generated code). Results for RoboTwin 1.0 and 2.0 are reported in Table 1 under *Vanilla* (one-shot generation), *FB* (feedback-based repair using execution logs), and *MM FB* (multimodal feedback with vision–language diagnostics). Per-task success rates are provided in Appendix 11.

Table 1: **Overall performance on tasks shared by RoboTwin 1.0 and 2.0.** Per-task success rates are in Appendix 11.

| Method | ASR | Top5-ASR | CR-Iter | Token |
|---|---|---|---|---|
| R1.0 Vanilla | 47.4% | 57.6% | 1.00 | 1236.6 |
| R1.0 + FB | 60.4% | 71.4% | 2.46 | 1190.4 |
| R1.0 + MM FB | 63.9% | 74.2% | 2.42 | 1465.0 |
| R2.0 Vanilla | 62.1% | 68.0% | 1.00 | **569.4** |
| R2.0 + FB | 66.7% | 73.6% | 1.89 | 581.6 |
| R2.0 + MM FB | **71.3%** | **78.6%** | **1.76** | 839.7 |

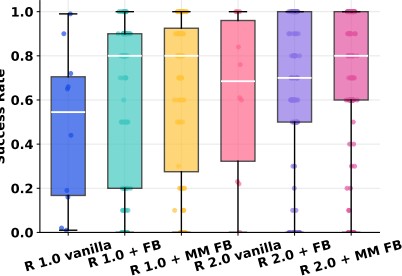

Figure 9: **Success Rate Distribution.**

Across all settings, multimodal feedback improves performance. In RoboTwin 1.0, ASR increases from 47.4% (Vanilla) to 63.9% (MM FB); in RoboTwin 2.0, from 62.1% to 71.3%. Top5-ASR also rises, indicating disproportionate gains for the best candidate programs. RoboTwin 2.0 converges faster than 1.0 (CR-Iter 1.76 vs. 2.42 under MM FB) and reduces token usage, especially in Vanilla (569.4 vs. 1236.6), reflecting more concise initial code. Figure 9 further shows that feedback narrows the success-rate distribution and raises the median; with multimodal feedback, RoboTwin 2.0 exhibits a compact distribution centered above 80%. Overall, three findings emerge: (1) vision–language feedback not only detects failures but also guides precise repairs; (2) architectural improvements in RoboTwin 2.0 accelerate convergence and reduce token usage; and (3) combining symbolic execution logs with perceptual diagnostics yields more reliable, semantically aligned expert data. Together, these results validate the effectiveness of our closed-loop, self-improving code generation architecture. Detailed setups, metric definitions, and additional analyses are provided in Appendix A.10.

## 4.2 RoboTwin 2.0 Benchmark

We present the RoboTwin 2.0 Benchmark for evaluating policy performance. Results on 50 RoboTwin tasks are reported in Appendix A.19, and Tab. 2 summarizes the average performance of RGB-based policies across evaluation settings. To assess generalization, we evaluate all 50 tasks on the Aloha–AgileX dual-arm platform. For each task, we train on 50 clean expert demonstrations and test with 100 rollouts under two conditions: *Easy* (no domain randomization) and *Hard* (domain randomization with clutter, lighting, texture, and height variation). We report success rate as the metric of few-shot adaptability and robustness. Appendix A.13 visualizes the benchmark setup, and Appendix A.5 details all training protocols.

Table 2: **Average Result of RoboTwin 2.0 Benchmark**. Full results are in Appendix A.19.

| Simulation Tasks | RDT | | Pi0 | | ACT | | DP | | DP3 | |
|---|---|---|---|---|---|---|---|---|---|---|
| | Easy | Hard | Easy | Hard | Easy | Hard | Easy | Hard | Easy | Hard |
| *Average (in %)* | 34.5 | 13.7 | 46.4 | **16.3** | 29.7 | 1.7 | 28.0 | 0.6 | **55.2** | 5.0 |

As shown in Tab. 2, under the Easy condition, ACT and DP perform substantially worse than the pretrained models RDT and Pi0 (29.7%, 28.0% vs. 34.5%, 46.4%), indicating that vision–language–action pretraining supplies strong priors that enable rapid policy learning from 50 demonstrations. Compared with RGB-based policies, DP3 attains the best few-shot performance in Easy (55.2%), highlighting the contribution of 3D information; however, its high success rate is partly attributable to idealized simulated depth and clean background segmentation. From the clean to the randomized Hard setting, all methods degrade: the non-pretrained models ACT, DP, and DP3 drop to 1.7%, 0.6%, and 5.0%, respectively, whereas RDT and Pi0 remain higher at 13.7% and 16.3%. These results indicate that vision–language–action pretraining provides useful priors for scene generalization and improves robustness to environmental variation, yet robustness under domain shift remains a central challenge. In conjunction with Secs. 4.3 and 4.4, these findings underscore the value of RoboTwin 2.0 as both a complementary dataset and a benchmark for systematic evaluation.

## 4.3 Assessing the Impact of RoboTwin 2.0 on Policy Robustness

We evaluate whether domain-randomized data in RoboTwin 2.0 enhances robustness to environmental perturbations. RDT and Pi0 are pre-trained on 9,600 expert trajectories drawn from 32 tasks (300 per task) under clean and domain-randomized settings. Off-the-shelf pretrained RDT and Pi0 are included as reference models without further fine-tuning. Generalization is examined on five unseen tasks using 50 clean demonstrations per task for single-task training and subsequent fine-tuning. ACT, DP, RDT, and Pi0 are then evaluated under domain-randomized conditions in previously unseen environments to quantify robustness. Detailed configurations are provided in Appendix A.4 and A.5.

As shown in Table 3, fine-tuning on clean data yields negligible gains in average success rate relative to pretrained baselines, indicating that non-randomized data do not improve robustness to environmental variation. This further suggests that the low simulated performance of pretrained VLA models is not attributable to a real-to-sim gap, since adding clean simulated data produces no clear benefit. In contrast, pretraining with RoboTwin 2.0 data substantially improves generalization: RDT and Pi0 attain relative gains of 31.9% and 29.3%, respectively. Notably, these gains persist even when

Table 3: **Evaluating the Impact of RoboTwin 2.0 on Policy Robustness.**

| Simulation Tasks | ACT | DP | RDT | RDT +Clean | RDT +Rand. | Pi0 | Pi0 +Clean | Pi0 +Rand. |
|---|---|---|---|---|---|---|---|---|
| Stack Bowls Two | 0.0% | 0.0% | 30.0% | 8.0% | **49.0%** | 41.0% | 55.0% | **62.0%** |
| Pick Dual Bottles | 0.0% | 0.0% | 13.0% | 12.0% | **17.0%** | 12.0% | **15.0%** | 7.0% |
| Move Can Pot | 4.0% | 0.0% | 12.0% | 13.0% | **18.0%** | 21.0% | **35.0%** | 22.0% |
| Place Object Basket | 0.0% | 0.0% | **17.0%** | 9.0% | 6.0% | 2.0% | 8.0% | **22.0%** |
| Place Shoe | 0.0% | 0.0% | 7.0% | 9.0% | **30.0%** | 6.0% | 6.0% | **18.0%** |
| Open Laptop | 0.0% | 0.0% | 32.0% | 21.0% | **35.0%** | 46.0% | 33.0% | **50.0%** |
| Press Stapler | 6.0% | 0.0% | 24.0% | 21.0% | **27.0%** | 29.0% | 26.0% | **31.0%** |
| Turn Switch | 2.0% | 1.0% | 15.0% | **24.0%** | 16.0% | **23.0%** | 21.0% | 21.0% |
| *Average* | 2.0% | 0.1% | 18.8% | 14.6% | 24.8% | 22.5% | 24.9% | 29.1% |

downstream training uses only clean, non-randomized data, demonstrating that domain-randomized pretraining with RoboTwin 2.0 confers robustness to visual and spatial variation. Consequently, models pretrained with RoboTwin 2.0 adapt to new tasks without additional augmentation or complex scene variation.

## 4.4 EVALUATION ON SIM-TO-REAL PERFORMANCE

To assess RoboTwin 2.0's impact on real-world robustness, we evaluate four bimanual tasks: *Stack Bowls*, *Handover Block*, *Pick Bottle*, and *Click Bell*. All experiments use RDT as the policy backbone on the COBOT-Magic dual-arm platform. We compare three training regimes: (1) 10 real-world demonstrations collected in clean tabletop environments; (2) the same demonstrations augmented with 1,000 domain-randomized synthetic trajectories generated under clutter, varied lighting, and diverse backgrounds; and (3) a synthetic-only model trained on the 1,000 synthetic trajectories. To improve robustness to camera jitter and calibration error, we apply random 3D perturbations to simulated camera poses (position and orientation), with translation magnitude bounded by 1 cm. We evaluate under four configurations: clean vs. cluttered tabletops crossed with seen vs. unseen backgrounds (Fig. 10). The synthetic-only model excludes seen backgrounds during training, so the corresponding entries in Table 4 are omitted. This setup tests whether RoboTwin 2.0 supports robust generalization without additional real-world data from visually complex scenes.

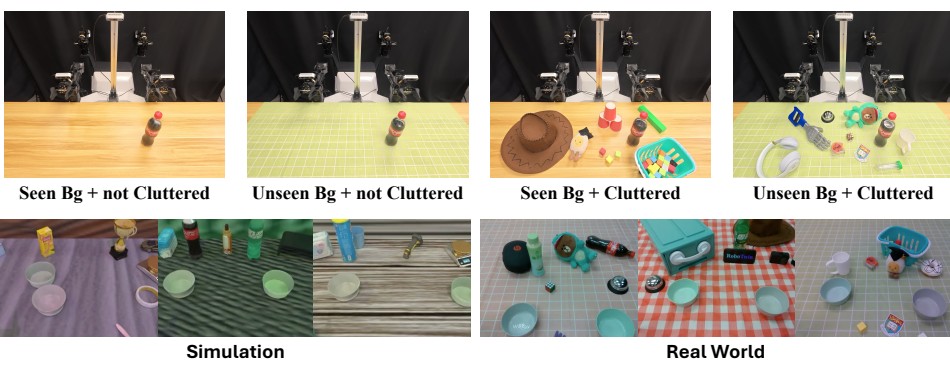

**Seen Bg + not Cluttered**   **Unseen Bg + not Cluttered**   **Seen Bg + Cluttered**   **Unseen Bg + Cluttered**

**Simulation**   **Real World**

Figure 10: **Real-World Evaluation Configurations and Sim-Real Comparison.**

RoboTwin 2.0 augmentation yields substantial robustness improvements in real-world bimanual policies. In the few-shot setting, which combines 1,000 domain-randomized synthetic trajectories with 10 real demonstrations, the average success rate across all evaluation configurations increases by 24.4%, with per-configuration gains of 13.5%, 27.5%, 23.5%, and 33.0%. In the zero-shot setting trained solely on synthetic data, the two unseen-background configurations improve by 21.0% and 20.5%. These gains are larger in visually complex scenes, indicating particular effectiveness under

Table 4: **Real-World Experiment Results.** We conduct controlled experiments on 4 dual-arm tasks: *Stack Bowls*, *Handover Block*, *Pick Bottle*, and *Click Bell*, each evaluated under 4 different settings.

| Real World Task | Background Type | Cluttered or Not | 10 Clean Real | 10 Clean Real + 1k RoboTwin 2.0 | 1k RoboTwin 2.0 (Zero-shot) |
|---|---|---|---|---|---|
| *Stack Bowls* | Seen | False | 22.0% | **64.0%** | / |
| | | True | 12.0% | **58.0%** | / |
| | Unseen | False | 10.0% | 50.0% | **60.0%** |
| | | True | 12.0% | **56.0%** | 52.0% |
| *Handover Block* | Seen | False | 40.0% | **48.0%** | / |
| | | True | **16.0%** | 12.0% | / |
| | Unseen | False | 36.0% | **56.0%** | 56.0% |
| | | True | 0.0% | **36.0%** | 20.0% |
| *Pick Bottle* | Seen | False | 20.0% | **36.0%** | / |
| | | True | 8.0% | **40.0%** | / |
| | Unseen | False | 4.0% | **26.0%** | 10.0% |
| | | True | 8.0% | 28.0% | **32.0%** |
| *Click Bell* | Seen | False | **36.0%** | 24.0% | / |
| | | True | 20.0% | **56.0%** | / |
| | Unseen | False | 12.0% | **24.0%** | 20.0% |
| | | True | 16.0% | **48.0%** | 14.0% |
| *Average* | Seen | False | 29.5% | **43.0%**$_{+13.5\%}$ | / |
| | | True | 14.0% | **41.5%**$_{+27.5\%}$ | / |
| | Unseen | False | 15.5% | **39.0%**$_{+23.5\%}$ | 36.5%$_{+21.0\%}$ |
| | | True | 9.0% | **42.0%**$_{+33.0\%}$ | 29.5%$_{+20.5\%}$ |

challenging conditions. We attribute the improvements to three factors: (1) the high visual and physical fidelity of RoboTwin 2.0, which enables direct sim-to-real transfer; (2) domain-randomized synthetic data that conditions policies on environmental variations absent from clean real-world demonstrations; and (3) large-scale simulation-based randomization that increases scene diversity and strengthens cross-scene transfer. Overall, these results suggest that a small amount of real-world data, when combined with rich RoboTwin 2.0 simulation, is sufficient to substantially narrow the sim-to-real gap.

Table 5: **Real robot performance with different sim–real mixtures.**

| | Click Bell | Place Empty Cup | Stack Bowls Two | Average |
|---|---|---|---|---|
| 50 real | 15.0% | 10.0% | 0.0% | 8.3% |
| 300 sim + 0 real | 35.0% | 10.0% | 0.0% | 15.0% |
| 300 sim + 10 real | 40.0% | 25.0% | 10.0% | 25.0% |
| 300 sim + 30 real | 55.0% | 35.0% | 20.0% | 36.7% |
| 300 sim + 50 real | 65.0% | 50.0% | 25.0% | 46.7% |

Beyond the above RDT-based dual-arm experiments, we further study how RoboTwin 2.0 interacts with larger amounts of real data and a different policy backbone. Specifically, we vary the ratio of real-world and simulated demonstrations and evaluate the resulting pi0 policies on real robots. On three tasks (*Click Bell*, *Place Empty Cup*, *Stack Bowls Two*), we compare: (i) 50 real demonstrations collected in a fixed scene; (ii) 300 domain-randomized simulated demonstrations; and (iii) mixtures of 300 simulated demonstrations with 0, 10, 30, or 50 real demonstrations. Each model is evaluated over 20 trials per task in unseen real scenes. As shown in Table 5, simulation-only training already outperforms real-only training, and adding modest amounts of real data on top of RoboTwin 2.0 simulation leads to consistent gains, reaching 46.7% average success with 300 sim + 50 real demonstrations. Taken together with the RDT results, these findings indicate that RoboTwin 2.0 not only enables strong zero-shot transfer, but also continues to provide sizeable benefits in more data-rich real-world regimes and across different policy architectures.

## 5 RELATED WORKS

### 5.1 DATASETS AND BENCHMARKS FOR ROBOTIC MANIPULATION

Physics-based simulators underpin much of manipulation research. SAPIEN Xiang et al. (2020) supports dynamic interaction with 2,300+ articulated objects, and ManiSkill2 Gu et al. (2023) provides millions of demonstrations. Meta-World Yu et al. (2020), CALVIN Mees et al. (2022), LIBERO Liu et al. (2023), and RoboVerse Geng et al. (2025) target multi-task, language-conditioned, lifelong, or domain-randomized settings, while RoboCasa Nasiriany et al. (2024) offers large-scale human demonstrations but lacks automation and a dual-arm focus. On the real-world side, large datasets such as AgiBot World Bu et al. (2025), RoboMIND Wu et al. (2024), Open X-Embodiment O'Neill et al. (2024), and Bridge Ebert et al. (2021) bridge sim-to-real with millions of trajectories.

Building on RoboTwin-1.0 Mu et al. (2025), RoboTwin 2.0 integrates LLM-driven feedback and systematic domain randomization over visual, physical, and task factors, yielding richer corpora and stronger generalization (Appendix A.3). Compared with prior OOD manipulation benchmarks such as GEMBench Garcia et al. (2025) and The Colosseum Pumacay et al. (2024), RoboTwin 2.0 provides much larger scale and richer domain randomization, with more than 700 assets, over 11k background textures, and 50 bimanual tasks across five embodiments. It extends domain randomization from attribute-level and evaluation-time perturbations to large-scale visual and geometric diversity, cluttered scene composition, and diverse language instructions that are applied consistently during both training and evaluation.

### 5.2 ROBOT LEARNING IN MANIPULATION

Task-specific policies Wang et al. (2024); Ke et al. (2024); Ze et al. (2024); Chi et al. (2023); Fu et al. (2024); Chen et al. (2025a); Liang et al. (2025; 2023; 2024); Wen et al. (2025b;a); Chen et al. (2025b) excel on individual tasks yet transfer poorly across embodiments. Foundation models trained on million-scale, multi-robot data generalize better: RT-1 Brohan et al. (2022) unifies vision, language, and action; RT-2 Brohan et al. (2023) co-fine-tunes vision–language models on web and robot data for semantic planning; RDT-1B Liu et al. (2024) and $\pi_0$ Black et al. (2024) use $> 1M$ episodes to capture diverse bimanual dynamics. OpenVLA Kim et al. and CogACT Li et al. (2024), with Octo Team et al. (2024), LAPA Ye et al., and OpenVLA-OFT Kim et al. (2025), demonstrate efficient adaptation to new robots and sensors. We contribute digital-twin data collection and broad domain randomization to produce realistic datasets that support robust, generalizable bimanual policies.

## 6 CONCLUSION

This paper introduced RoboTwin 2.0, a scalable simulation framework for generating diverse, high-fidelity expert data for robust bimanual manipulation. The system integrates MLLM-based expert code generation, embodiment-adaptive behavior synthesis, and comprehensive domain randomization, addressing key limitations of prior synthetic data generators. Leveraging an annotated object library and automated trajectory synthesis, RoboTwin 2.0 produces visually, linguistically, and physically rich datasets while reducing manual effort. Experiments demonstrate consistent improvements in cluttered scenes, enhanced generalization to unseen tasks, and reliable cross-embodiment transfer; notably, few-shot and zero-shot evaluations indicate measurable sim-to-real improvements, showing that domain-randomized, semantically grounded synthetic data can substantially reduce real-world data requirements. To support the community, we release as open source RoboTwin-OD, a pre-collected trajectory dataset, a standardized benchmark, and a scalable data-collection toolchain. RoboTwin 2.0 provides a principled basis for unified benchmarking and scalable sim-to-real pipelines.

## ETHICS STATEMENT

Our study investigates simulation-based data generation and policy learning for robotic manipulation. Experiments were conducted in simulation and in controlled laboratory settings without human subjects, personally identifiable information, or sensitive biometric data; therefore, institutional review (IRB) was not required. All physical experiments followed standard lab safety protocols

(e.g., emergency stop, workspace clearance, tool guards). We release assets and code under research-friendly licenses and comply with third-party licenses for simulators and object models. To reduce representational bias, our benchmark includes diverse scenes, clutter levels, lighting conditions, and multi-embodiment tasks; nevertheless, residual biases may persist (e.g., category coverage, tabletop assumptions). Potential dual-use risks—such as automating unsafe behaviors—are mitigated by (1) restricting release to research use, (2) documenting known failure modes, and (3) providing safety guidelines for real-robot deployment. We disclose compute usage and hardware in the appendix to support environmental transparency. The authors report no conflicts of interest and adhered to the ICLR Code of Ethics throughout submission, reviewing interactions, and discussion.

## REPRODUCIBILITY STATEMENT

We take several steps to enable reproducibility. The paper specifies model and training details for all baselines (ACT, DP, DP3, RDT, Pi0) and our methods, including architectures, hyperparameters, and optimization settings (see Sec. 2 and Sec. 4). We fix random seeds and specify hardware/software stacks (CUDA/driver, simulator version) to minimize variance; instructions for reproducing tables and figures from scratch are provided in a README.

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

# A APPENDIX

## A.1 THE USE OF LARGE LANGUAGE MODELS

No large language models or AI-assisted tools were used at any stage of this work, including writing, coding, data processing, analysis, figure generation, or conclusions. All text and code were authored, reviewed, verified, and tested solely by the authors, who take full responsibility for the content and any errors.

## A.2 EVALUATING EFFICIENCY WITH AND WITHOUT ADAPTIVE GRASPING

Table 6: **Overall Performance Comparison between RoboTwin 1.0 and RoboTwin 2.0.**

| Method | Aloha-AgileX | Piper | Franka | UR5 | ARX-X5 | Average |
|---|---|---|---|---|---|---|
| RoboTwin 1.0 | 65.1% | 2.4% | **67.3%** | **57.6%** | 68.6% | 52.2% |
| RoboTwin 2.0 | **78.8%** | **25.1%** | 67.2% | 57.1% | **74.2%** | **60.5%** |
| *Difference* | **+13.7%** | **+22.7%** | **-0.1%** | **-0.5%** | **+5.6%** | **+8.3%** |

We evaluate embodiment-aware grasp augmentation by measuring automated data-collection success on 50 RoboTwin 2.0 tasks across five robot embodiments. As shown in Table 6, our pipeline outperforms the RoboTwin 1.0 baseline, which lacks diverse grasping and candidate augmentation, with an average gain of 8.3%. Benefits are concentrated on lower-DoF platforms: Aloha-AgileX, Piper, and ARX-X5 improve by 13.7%, 22.7%, and 5.6%, respectively. High-DoF arms with large reachable workspaces, such as Franka and UR5 (7-DoF), show little change, consistent with sufficient kinematic flexibility. These results indicate that augmentation supplies additional feasible grasps that mitigate planning constraints on low-DoF manipulators. Full results are reported in Appendix A.20.

## A.3 BENCHMARKING ROBOTWIN 2.0 AGAINST EXISTING DATASETS

We compare RoboTwin 2.0 against existing benchmarks and datasets across several key dimensions, including the number of supported tasks, the presence of domain randomization, support for automatic data generation, and compatibility with vision-language-action (VLA) model training and evaluation. The comparison is summarized in Table 7.

Table 7: **Comparison of RoboTwin 2.0 with previous manipulation benchmarks and datasets.**

| Benchmark & Dataset | #Tasks | Domain Randomization | Auto Data Generation | VLA Model Train & Eval |
|---|---|---|---|---|
| Meta-world Yu et al. (2020) | 50 | ✗ | ✓ | ✗ |
| Robosuite Zhu et al. (2020) | 9 | ✗ | ✗ | ✗ |
| RoboCasa Yu et al. (2020) | 25 | ✓ | ✗ | ✗ |
| Maniskill2 Gu et al. (2023) | 20 | ✗ | ✓ | ✗ |
| AutoBio Lan et al. (2025) | 16 | ✗ | ✓ | ✓ |
| RoboTwin 1.0 Mu et al. (2025) | 14 | ✗ | ✓ | ✓ |
| RoboTwin 2.0 (ours) | 50 | ✓ | ✓ | ✓ |

## A.4 DOMAIN RANDOMIZATION SETTING

Domain randomization in all experiments includes cluttered scenes, random lighting, table height variation (up to 3 cm), unseen language instructions and randomized background textures.

## A.5 POLICIES TRAINING DETAILS

We adopt joint angles as the model's prediction target, formulating action prediction as joint-angle regression.

**RDT** in experiment 4.3 was finetuned for 100,000 steps with a batch size of 16 per GPU on 8 GPUs, and all single-task fine-tuning was conducted for 10,000 steps with a batch size of 16 per GPU on 4 GPUs. In all cases, we initialize from the publicly released RDT pretrained weights.

**Pi0** in experiment 4.3 was pretrained for 100,000 steps with a batch size of 32, and all fine-tuning was performed for 30,000 steps using the same batch size with LoRA-based fine-tuning. In all cases, we initialize from the publicly released Pi0 pretrained weights.

**ACT** was trained under a unified setup with a chunk size of 50, batch size of 8, and single-GPU training for 6,000 epochs. During deployment, we applied `temporal_agg` for temporal aggregation to improve execution stability.

**DP** was trained for 600 epochs with a batch size of 128 and a planning horizon of 8.

**DP3** was trained for 3,000 epochs with a batch size of 256, using a planning horizon of 8 and a point cloud resolution of 1,024, with precise segmentation of the background and tabletop.

### A.6  ABLATION ON DOMAIN RANDOMIZATION FACTORS

To understand which domain randomization factors contribute most to policy robustness, we conduct an ablation study and report the results in Table 8. For each factor (background, clutter, table height, lighting), we disable only that factor's randomization, collect 100 trajectories for training, and then evaluate the resulting policy under the full domain randomized setting. This isolates the contribution of each factor while keeping all other conditions fixed. The table reports success rates (in %) for four representative tasks, with ACT and DP shown as "ACT / DP" in each cell. Disabling background or clutter randomization leads to the largest performance drops, while turning off height or lighting randomization results in smaller but still noticeable degradation. When all randomization factors are disabled, the success rate becomes very low, indicating that our visual domain randomization, especially in background and clutter, is a primary driver of learning robust behaviors.

Table 8: **Ablation on domain randomization factors.** Each entry shows success rate (in %) for ACT / DP. "BG" denotes background randomization.

| Task | BG↓ | Clutter↓ | Height↓ | Light↓ | All Rand.↓ |
|------|------|----------|---------|--------|------------|
| Adjust Bottle | 50 / 49 | 64 / 91 | 94 / 89 | 95 / 98 | 23 / 0 |
| Beat Block Hammer | 3 / 2 | 4 / 39 | 3 / 23 | 7 / 65 | 3 / 0 |
| Move Can Pot | 31 / 4 | 28 / 29 | 53 / 37 | 41 / 34 | 4 / 0 |
| Stack Bowls Two | 14 / 3 | 35 / 64 | 29 / 60 | 36 / 81 | 0 / 0 |
| **Average** | 24.5 / 14.5 | 32.8 / 55.8 | 44.8 / 52.3 | 44.8 / 69.5 | 7.5 / 0 |

### A.7  ROBUSTNESS UNDER DYNAMIC SCENE CHANGES

In Sec. 4.4, our notion of robustness refers to robustness to substantial variations in scene layout and background appearance, rather than arbitrary external disturbances. To make this explicit, we conduct additional real robot experiments that focus on dynamically changing scenes. On three real world tasks with pi0, we train two policies: one using 50 real demonstrations collected in a single fixed scene ("50 real"), and one using 300 domain randomized simulated demonstrations from RoboTwin 2.0 combined with the same 50 real demonstrations ("300 sim + 50 real"). During evaluation, we deliberately introduce dynamic perturbations by randomly moving scene objects and changing the tabletop background between episodes. As shown in Table 9, the policy trained only on fixed scene real data completely fails in this setting (0.0% success on all three tasks), whereas the policy trained with RoboTwin 2.0 augmented data maintains a 26.7% average success rate. This large gap under dynamic scene changes supports our claim that RoboTwin 2.0 significantly improves policy robustness to realistic visual and spatial variations.

### A.8  SUPPORT FOR FLEXIBLE EMBODIMENT COMBINATIONS

Our object-centric, embodiment-agnostic data generation framework enables seamless deployment across a wide range of dual-arm robotic systems. The pipeline supports flexible embodiment configurations, allowing arbitrary combinations of heterogeneous manipulators and relative arm placements.

Table 9: **Real robot performance under dynamic scene changes.** Success rate in %.

|  | Click Bell | Place Empty Cup | Stack Bowls Two | Average |
|---|---|---|---|---|
| 50 real | 0.0% | 0.0% | 0.0% | 0.0% |
| 300 sim + 50 real | 40.0% | 30.0% | 10.0% | 26.7% |

This design ensures compatibility with diverse hardware setups and facilitates extensibility to future robotic platforms.

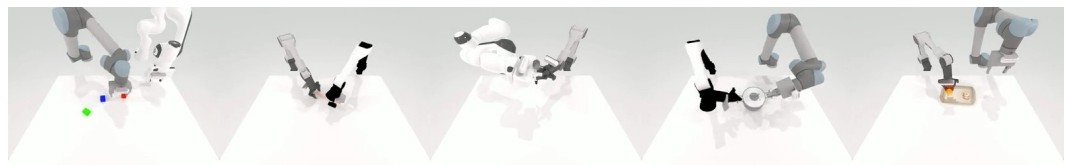

Figure 11: **Heterogeneous Dual-Arm Control via Object-Centric Manipulation.**

To execute high-success-rate manipulation trajectories across different embodiments (see Section 2.3), we integrate Curobo, a high-performance, GPU-accelerated motion planner that enables efficient and reliable planning under varied kinematic constraints.

Currently, our framework supports five robotic arms—Franka, Piper, UR5, ARX-X5, and Aloha-AgileX—along with multiple gripper types, including the Panda gripper and WSG gripper. As shown in Fig. 11, we demonstrate successful task executions across a variety of dual-arm pairings, highlighting RoboTwin 2.0's ability to scale to heterogeneous robot configurations and its readiness for future real-world deployment.

## A.9 IMPROVEMENTS OF ROBOTWIN 2.0 OVER ROBOTWIN 1.0 POLICY CODEBASE

| Metric | RoboTwin 1.0 | RoboTwin 2.0 |
|---|---|---|
| Prompt Token Length ↓ | 5901.0 | **4719.1** |
| Code Token Length ↓ | 1236.6 | **569.4** |
| Parallelism Control ↑ | ✗ | ✓ |
| AST Similarity Wen et al. (2019) ↑ | 23.72% | **44.78%** |
| CodeBLEU Similarity Ren et al. (2020) ↑ | 17.18% | **18.53%** |
| CodeBERT Similarity Feng et al. (2020) ↑ | 97.72% | **98.80%** |
| Unixcoder Similarity Guo et al. (2022) ↑ | 76.24% | **82.21%** |
| Avg. VLM Token Cost (per observation) | – | 6894 |

Table 10: **Code Generation Efficiency and Quality Comparison.** Evaluation of prompt and generated code characteristics, along with code similarity metrics (AST Structural Similarity, CodeBERT, Unixcoder cosine similarity) against expert-written code, for RoboTwin 1.0 and RoboTwin 2.0 in zero-shot generation. The VLM observer cost is also reported for RoboTwin 2.0.

We first quantify the architectural impact of RoboTwin 2.0 in a one-shot generation without code repair and iterative refinement. Table 10 shows that RoboTwin 2.0 yields significantly shorter programs (569.4 vs. 1236.6 tokens), with reduced prompt length and higher structural similarity to human-written code. Crucially, it enables dual-arm parallelism via a unified API abstraction, which is absent in RoboTwin 1.0.

These improvements stem from the structured prompting and geometric API modularization designed into RoboTwin 2.0. Higher AST similarity (+21.06%), CodeBERT similarity (+1.08%), and Unixcoder alignment (+5.97%) indicate that RoboTwin 2.0 not only reduces code size but also improves semantic clarity and functional alignment.

In addition, RoboTwin 2.0 integrates a **VLM observer**, a plug-and-play module triggered only when execution fails. To quantify its overhead, we estimated VLM usage via the Kimi API (assuming each image = 1,024 tokens) over three representative tasks: the average cost was 6,295 input tokens and

599 output tokens, totaling **6,894 tokens**. While this introduces moderate overhead, the VLM enables RoboTwin 2.0 to catch and correct errors invisible to execution logging, significantly enhancing robustness and overall task success. Importantly, the observer remains optional and can be disabled when prioritizing token efficiency.

## A.10 Experimental Details and Metric Definitions for Code Generation

We use the *DeepSeek-V3* model for program synthesis and the *moonshot-v1-32k-vision-preview* model for multimodal error localization and verification. These models were selected for their strong performance in language reasoning and visual understanding while maintaining efficiency suitable for large-scale iterative refinement. The success rate of the $i$-th program is computed as $R_i = \frac{1}{M} \sum_{j=1}^{M} s_{i,j}$, and the final success rate for a given task under a specific system variant is then defined as $R_{\text{task}} = \frac{1}{N} \sum_{i=1}^{N} R_i$.

### A.10.1 Metric Definitions

We report the following metrics across all tasks:

**ASR** is the average of $R\_task$ across all 10 tasks. It reflects overall task performance across all generated programs.

**Top5-ASR** is the mean success rate computed using only the top 5 highest-performing programs per task. This metric estimates system potential under a best-of-selection strategy.

**CR-Iter** indicates the average number of feedback iterations required per task before reaching a success rate above 50% or exhausting the iteration budget.

**Token** denotes the average number of tokens of policy code generated by the language model per task. It serves as a proxy for computational cost and LLM inference budget.

These metrics jointly evaluate both the reliability and efficiency of the expert data generation pipeline under varying conditions of feedback, model capability, and refinement strategy.

### A.10.2 Task-Specific Performance Comparison on Code Generation

We compare the code generation success rates of RoboTwin 2.0 and RoboTwin 1.0 across all tasks. As shown, RoboTwin 2.0 consistently matches or outperforms the baseline on the majority of tasks, demonstrating the effectiveness of our multimodal feedback and refinement pipeline.

Table 11: Task-Specific Performance Comparison between RoboTwin 2.0 and RoboTwin 1.0. R1.0/R2.0: RoboTwin 1.0 / 2.0. Bold numbers indicate the best result for each task.

| Task | R1.0 Vanilla | R1.0 + FB | R1.0 + MM FB | R2.0 Vanilla | R2.0 + FB | R2.0 + MMFB |
|---|---|---|---|---|---|---|
| beat_block_hammer | 16% | 48% | **56%** | 23% | 34% | 53% |
| handover_block | 2% | 41% | 45% | 17% | **50%** | 27% |
| pick_diverse_bottles | **65%** | **65%** | 64% | 60% | 60% | 62% |
| pick_dual_bottles | 99% | 99% | **100%** | **100%** | **100%** | **100%** |
| place_container_plate | 66% | 79% | **91%** | 84% | 84% | 82% |
| place_dual_shoes | 19% | 22% | **25%** | 0% | 2% | 22% |
| place_empty_cup | 90% | 90% | **100%** | 61% | 61% | 85% |
| place_shoe | 72% | 90% | 90% | **100%** | **100%** | **100%** |
| stack_blocks_three | 1% | 2% | 4% | 76% | 76% | **82%** |
| stack_blocks_two | 44% | 68% | 64% | **100%** | **100%** | **100%** |

### A.10.3 Per-task Success Rates of Code Generation

We report the success rates of all tasks in Tab. 12.

Table 12: **Per-task success rates of our proposed R2.0 + MM FB algorithm on all RoboTwin 2.0-supported tasks.**

| Task | Rate | Task | Rate | Task | Rate | Task | Rate |
|---|---|---|---|---|---|---|---|
| Adjust Bottle | 100% | Beat Block Hammer | 53% | Blocks Ranking Rgb | 80% | Blocks Ranking Size | 80% |
| Click Alarmclock | 0% | Click Bell | 10% | Dump Bin Bigbin | 0% | Grab Roller | 74% |
| Handover Block | 27% | Handover Mic | 0% | Hanging Mug | 0% | Lift Pot | 40% |
| Move Can Pot | 30% | Move Pillbottle Pad | 50% | Move Playingcard Away | 90% | Move Stapler Pad | 100% |
| Open Laptop | 0% | Open Microwave | 0% | Pick Diverse Bottles | 62% | Pick Dual Bottles | 100% |
| Place A2B Left | 50% | Place A2B Right | 60% | Place Bread Basket | 0% | Place Bread Skillet | 0% |
| Place Can Basket | 0% | Place Cans Plasticbox | 100% | Place Container Plate | 82% | Place Dual Shoes | 22% |
| Place Empty Cup | 85% | Place Fan | 70% | Place Burger Fries | 100% | Place Mouse Pad | 100% |
| Place Object Basket | 0% | Place Object Scale | 80% | Place Object Stand | 90% | Place Phone Stand | 0% |
| Place Shoe | 100% | Press Stapler | 0% | Put Bottles Dustbin | 0% | Put Object Cabinet | 0% |
| Rotate Qrcode | 80% | Scan Object | 0% | Shake Bottle | 0% | Shake Bottle Horizontally | 0% |
| Stack Blocks Three | 82% | Stack Blocks Two | 100% | Stack Bowls Three | 20% | Stack Bowls Two | 30% |
| Stamp Seal | 20% | Turn Switch | 0% | *Avg Success Rate* | *43.34%* | | |

### A.10.4 MULTIMODAL OBSERVATION AND ERROR LOCALIZATION

To further investigate the capability of the VLM observer, we manually curated a dataset of 130 execution sequences, including 101 failed trials and 29 successful trials. Each sequence consists of the natural language task instruction, a series of visual observations, and policy code. This dataset enables us to evaluate both binary error detection and fine-grained error localization.

**Error Detection.** The VLM observer was first tasked with evaluating whether a robotic execution successfully completed the instructed task. The confusion matrix is as follows: TP = 16, FP = 61, TN = 40, FN = 13. The derived performance metrics are: Accuracy = 0.431, Precision = 0.208, Recall = 0.552, and F1-score = 0.302. These results suggest that while the VLM observer is reasonably sensitive in identifying execution failures (recall), its tendency to over-predict errors leads to a high false positive rate.

**Error Localization.** For failed executions, we further tested whether the VLM observer could correctly localize the program step or code snippet that led to failure. Among the 40 true negative samples (correctly identified failed trials), the observer successfully pinpointed the failure step in 12 cases, yielding a localization accuracy of 30%. This indicates that, while the observer shows promising diagnostic ability, there remains significant room for improvement in reliable error attribution.

**Case Study.** We illustrate one successful case of VLM observer reasoning. Given a task instruction and five visual observations, the observer produced a step-by-step analysis:

Step 1: The left arm successfully picks up the mug from the left side.
Step 2: The left arm successfully places the mug at the middle position.
Step 3: The right arm successfully picks up the mug from the middle position.
Step 4: The right arm attempts to hang the mug on the rack but fails.
Step 5: The right arm is in the process of moving away after attempting to hang the mug.
The overall task was not successfully completed. The failure occurred in Step 4, where the right arm attempted to hang the mug on the rack but did not succeed. The mug is not properly positioned on the rack. The error message in the code indicates an AttributeError: 'list' object has no attribute 'p'. This error occurs in the line where the target_point is being set to middle_target_pose.p. The issue is likely due to the fact that middle_target_pose is a list, and lists do not have an attribute 'p'. The code should be modified to correctly access the position data from the middle_target_pose list.
To fix the error, the code should be updated to access the position data correctly. For example, if middle_target_pose is a list containing the position and orientation, the position data should be accessed using middle_target_pose[0] instead of middle_target_pose.p. Additionally, the failure in Step 4 could be due to the incorrect position or orientation of the mug when attempting to hang it on the rack. The code should be reviewed to ensure that the mug is positioned and oriented correctly before executing the hanging motion.

**Failure Modes.** Despite these successes, we also observed cases where the VLM observer fails. Misclassifications primarily occur in failed executions: the observer may hallucinate correct outcomes

or overlook subtle orientation errors. For instance, in a stacking task where boxes are misaligned, the observer sometimes marks the execution as successful if all boxes are placed on top of each other. Similarly, in tasks requiring fine-grained pose accuracy (e.g., placing shoes in the correct orientation), small but critical errors may be ignored. Furthermore, failures stemming from invisible factors—such as incorrect grasp axis parameters—remain challenging for purely vision-based observers to diagnose.

### A.10.5 LLM-GENERATED CODE AND HUMAN-WRITTEN CODE CASE STUDY

**LLM-Generated Code (`gpt_place_shoe`)**

```
class gpt_place_shoe(place_shoe):
    def play_once(self):
        # Initial observation
        self.save_camera_images(task_name="place_shoe", step_name="
            step1_initial_scene_state", generate_num_id="generate_num_0")

        # Get the shoe's position to determine which arm to use
        shoe_pose = self.shoe.get_pose()
        shoe_position = shoe_pose.p
        arm_tag = ArmTag("left" if shoe_position[0] < 0 else "right")

        # Grasp the shoe
        self.move(self.grasp_actor(actor=self.shoe, arm_tag=arm_tag,
            pre_grasp_dis=0.1, grasp_dis=0))
        self.save_camera_images(task_name="place_shoe", step_name="
            step2_shoe_grasped", generate_num_id="generate_num_0")

        # Lift the shoe up to avoid collision
        self.move(self.move_by_displacement(arm_tag=arm_tag, z=0.07,
            move_axis='world'))
        self.save_camera_images(task_name="place_shoe", step_name="
            step3_shoe_lifted", generate_num_id="generate_num_0")

        # Get target pose from the top functional point of the target block
        target_pose = self.target_block.get_functional_point(1, "pose")

        # Place the shoe on the target block with head towards left side
        self.move(self.place_actor(actor=self.shoe, arm_tag=arm_tag,
            target_pose=target_pose, functional_point_id=0, pre_dis=0.1,
            dis=0.02, is_open=True, constrain="align", pre_dis_axis='fp'))
        self.save_camera_images(task_name="place_shoe", step_name="
            step4_shoe_placed", generate_num_id="generate_num_0")

        # Lift the gripper up after placing
        self.move(self.move_by_displacement(arm_tag=arm_tag, z=0.07,
            move_axis='world'))

        # Return arm to origin
        self.move(self.back_to_origin(arm_tag=arm_tag))

        # Final observation
        self.move(self.save_camera_images(task_name="place_shoe", step_name
            ="step5_final_scene_state", generate_num_id="generate_num_0"))
```

Listing 1: LLM-Generated Code for gpt_place_shoe Task

**Human-Written Code (`place_shoe`)**

```
class place_shoe(base_task):
    def play_once(self):
        # Get the shoe's position to determine which arm to use
        shoe_pose = self.shoe.get_pose().p
        arm_tag = ArmTag("left" if shoe_pose[0] < 0 else "right")
```

```
# Grasp the shoe with specified pre-grasp distance and gripper
    position
self.move(self.grasp_actor(self.shoe, arm_tag=arm_tag,
    pre_grasp_dis=0.1, gripper_pos=0))

# Lift the shoe up by 0.07 meters in z-direction
self.move(self.move_by_displacement(arm_tag=arm_tag, z=0.07))

# Get target_block's functional point as target pose
target_pose = self.target_block.get_functional_point(0)

# Place the shoe on the target_block with alignment constraint and
    specified pre-placement distance
self.move(self.place_actor(self.shoe, arm_tag=arm_tag, target_pose=
    target_pose, functional_point_id=0, pre_dis=0.12, constrain="
    align"))

# Open the gripper to release the shoe
self.move(self.open_gripper(arm_tag=arm_tag))
```

Listing 2: Human-Written Code for `place_shoe` Task

The LLM generated code tends to be more verbose, explicitly logging intermediate visual states and detailing parameters (e.g., `pre_dis_axis='fp'`, `is_open=True`), while human-written scripts are more minimal, omitting intermediate steps and favoring compact execution. Despite functional similarity, the structural differences illustrate that **MLLM-generated programs are not only executable but emphasize step-by-step clarity**, contributing to more robust feedback and repair.

## A.11 TASK INSTRUCTION AND OBJECT DESCRIPTION EXAMPLE

---
**Instruction Templates (task: 'Pick Dual Bottles')**

```
"Use {a} to place {A} left of {B}.", "Set {A} to the left of {B}.", "Move {A} beside
{B} using {a}.", "Place {A} on {B}'s left side.", "Using {a}, position {A} next to
{B}.", "Stick {A} on the left of {B}.", "Use {a} and place {A} on {B}'s left.", etc
```
---

---
**Object Description**

```
# object id - '001_bottle/0':
"red bottle", "red soda bottle", "plastic red bottle", "red bottle with yellow label",
"red plastic bottle with smooth surface", "yellow text printed on red bottle surface",
"red bottle with white label design and markings", "red bottle with white sealing and
brown top screw cap", etc
# object id - '039_mug/0':
"black mug", "dark coffee mug", "sleek black mug", "black ceramic mug", "single-handle
mug", "smooth black surface mug", "medium-sized drinking mug", "round mug with curved
side", "dark mug with sturdy handle", "solid black mug with smooth finish", etc
```
---

## A.12 PROMPTS FOR GENERATING TASK INSTRUCTIONS AND OBJECT DESCRIPTIONS

```
# Task Instruction Template
- Goal: Generate task instruction template
- Requirements:
  - Generate 60 items. Vary in sentence length and structure
  - Use natural action verbs (grab, slide, place)
- split
  - 50 items for training
  - 10 items for evaluation
```

```
## Schema Requirements
- Goal: Use placeholders for objects in instructions
- Requirements:
  - Format: {X} for objects defined in schema
  - Include all object placeholders ({A-Z}) in every instruction
  - Omit arm references and placeholders ({a-z}) in 50% of instructions
  - Ensure natural flow when placeholders are replaced with actual values

# Object Description
- Goal: Generate natural object descriptions for robotic manipulation
- Requirements:
  - Generate 15 items. Vary in sentence length and structure
  - Use natural oral language
  - Include essential physical properties (color, shape, size, texture)
  - Use noun-focused phrases
  - For multi-part objects, use structures like 'X with Y'
- split
  - 12 items for training
  - 3 items for evaluation

# Episode
An episode is a specified task, in which each task may have different
    objects to be manipulated,
resulting in the same task template being reused by replacing the
    placeholders with specific objects.
For example:
  {A} -> 'medium-sized yellow bottle'
  {A} -> 'green drink bottle with bold labels'

General Task -> Specific Episode:
  {A} -> bottle/0.glb
  {A} -> bottle/1.glb

The number of task instructions for an episode can be calculated by:
  Episode_num = TaskInstruction_num * ObjectDescription_num
```

Listing 3: Prompts for Generating Task Instructions and Object Descriptions

## A.13 VISUALIZATION OF ROBOTWIN 2.0 BENCHMARK SETTING

We visualize the simulation settings of the RoboTwin 2.0 benchmark in Fig. 12. All models are trained on 50 clean (non-randomized) demonstrations per task (blue). For evaluation, the *Easy* setting also uses clean environments, while the *Hard* setting employs domain-randomized environments (green).

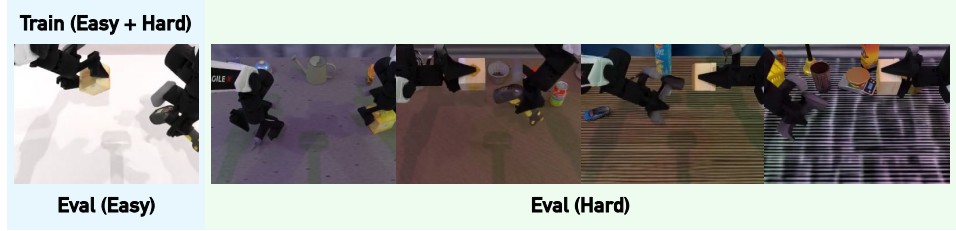

Figure 12: **Visualization of RoboTwin 2.0 Benchmark Settings.**

## A.14 VISUALIZATION OF RENDERED MESHES AND COLLISION SHAPES OF OBJECTS IN ROBOTWIN-OD

To illustrate how complex geometries are handled, we visualize both the rendered meshes and collision shapes for five representative objects in Fig. 14, including those with internal cavities such as mugs. Their holes and fine structural details are preserved in the final assets. Specifically, after

generating each object via AIGC, we perform convex decomposition in Blender and then merge the resulting parts to obtain smooth, physically stable collision bodies that are compatible with the SAPIEN simulator.

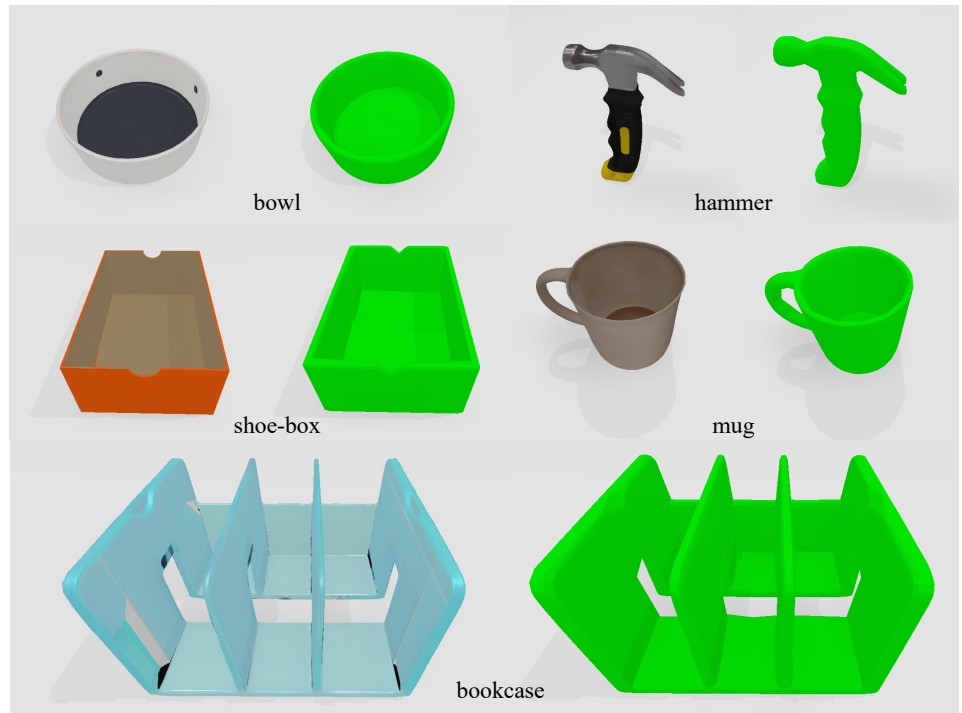

Figure 13: **Visualization of RoboTwin-OD Objects.**

### A.15 SAPIEN-IPC–BASED TACTILE DATA ACQUISITION SETUP

We build our simulated tactile data acquisition pipeline on top of the SAPIEN-IPC framework, which provides a flexible and scalable environment for generating high-fidelity contact interactions.

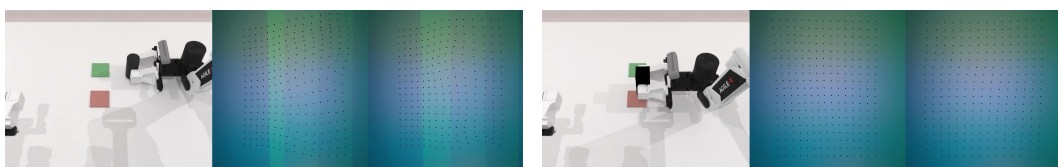

Figure 14: **Visualization of Tactile Data Acquisition.**

### A.16 USER STUDY ON INSTRUCTION NATURALNESS

To further evaluate the naturalness of our LLM-generated instructions, we conduct a user study on the 567 objects with textual descriptions in RoboTwin-OD. For each object, we first generate 15 candidate instructions using an LLM, as described in Sec. 3.1. We then ask 5 volunteers to write additional natural-language descriptions for each object, yielding in total 20 descriptions per object (15 LLM-generated and 5 human-written).

A separate group of 5 volunteers is then presented with the 20 descriptions for each object and asked to select the one that is *most likely* written by a human. Since 5 out of 20 descriptions are human-authored, the ground-truth proportion of human-written descriptions in the pool is 25%. Table 13 reports the success rate of each volunteer in correctly identifying a human-written description.

The average success rate is 23.8%, which is very close to the 25% ground-truth proportion. This near-chance performance indicates that human annotators have difficulty distinguishing between

Table 13: Success rate of volunteers in identifying human-written instructions among mixed pools of human- and LLM-generated descriptions. Chance level corresponds to 25%.

|                  | Vol. 1 | Vol. 2 | Vol. 3 | Vol. 4 | Vol. 5 | Average |
|------------------|--------|--------|--------|--------|--------|---------|
| Success rate (%) | 21.0   | 19.9   | 26.1   | 27.2   | 22.9   | 23.8    |

Table 14: **CLIP-based global distribution matching and coverage between our 11k background texture library and real-world textures from DTD.** We extract ViT-B/32 CLIP features for 11k textures from our library and 5,640 real textures from DTD. Coverage is measured as nearest-neighbor cosine similarity from each real texture to our texture pool.

|             | #Ours  | #Real | FID $\downarrow$ | mean $\uparrow$ | median $\uparrow$ | p10 $\uparrow$ | p90 $\uparrow$ | min / max     |
|-------------|--------|-------|-------|-------|--------|-------|-------|---------------|
| Ours vs. DTD | 11,000 | 5,640 | -0.36 | 0.839 | 0.848  | 0.749 | 0.914 | 0.544 / 0.979 |

human-written and LLM-generated instructions, suggesting that our instruction pool is linguistically close to natural human descriptions rather than being obviously synthetic or overly templated.

### A.17    CONTROL FREQUENCY AND CONTROLLER IMPLEMENTATION

RoboTwin 2.0 is built on the SAPIEN simulator, where each simulation step corresponds to 0.004 seconds of real time. In our default data collection setting, we record one sample every 15 simulation steps, which gives an effective sampling rate of approximately 16.67 Hz (this rate can be adjusted via configuration parameters). The temporal spacing of policy inputs and outputs therefore naturally aligns with this sampling interval.

For policy execution in both simulation and on real robots, we interpolate the predicted actions to match the underlying low level controller frequencies. When the policy outputs joint space actions, we use TOPP based interpolation to generate time parameterized joint trajectories; when the policy outputs end effector poses, we use trajectory planning to construct smooth Cartesian motion. In our setup, the controller runs at 250 Hz in simulation and at 30 Hz on the real robot, so the interpolated trajectories bridge the gap between the policy output rate ($\sim$ 16.67 Hz) and the higher frequency control loops.

### A.18    TEXTURE DISTRIBUTION VS. REAL-WORLD BACKGROUNDS

To evaluate whether our 11k background texture library reasonably approximates real-world background statistics, we perform an analysis in a pretrained CLIP feature space. Specifically, we extract ViT-B/32 CLIP Radford et al. (2021) embeddings (OpenAI weights) for two sets of backgrounds: (i) our 11k textures from the background library and (ii) 5,640 real textures from the Describable Textures Dataset (DTD) Cimpoi et al. (2014), a widely used real-world texture benchmark containing in-the-wild images annotated with human-describable attributes (e.g., "banded", "wrinkled", "cracked"). We then (1) approximate each set by a Gaussian in CLIP space and compute a CLIP-FID between our textures and DTD, and (2) evaluate coverage by, for each real (DTD) texture, finding its nearest neighbor in our pool and recording the cosine similarity between their CLIP embeddings.

As summarized in Tab. 14, the CLIP-FID between our texture library and DTD is essentially zero ($-0.36$, within numerical noise), indicating that the first- and second-order statistics of our textures are almost identical to those of real textures in this feature space. Moreover, the nearest-neighbor cosine similarity from DTD to our library exhibits a high mean (0.839) and median (0.848), with the 90th percentile reaching 0.914, meaning that at least 90% of real textures have a very close counterpart in our pool. Even the worst-case real sample still attains a cosine similarity of 0.544 to its closest neighbor, so we do not observe any real textures that are nearly orthogonal to our manifold in CLIP space. Overall, these results indicate that our curated texture library is not an artificial or overly narrow distribution, but instead provides a high-coverage, globally well-matched approximation to real-world background statistics.

## A.19 FULL ROBOTWIN 2.0 BENCHMARK

We report the evaluation results of five policies on the RoboTwin 2.0 benchmark under the *Easy* and *Hard* settings. Note that these two settings differ only in evaluation conditions, while the training setup remains identical.

Table 15: **RoboTwin 2.0 Simulation Benchmark (clean vs randomized, 50+ tasks).**

| Simulation Task | RDT | | Pi0 | | ACT | | DP | | DP3 | |
|---|---|---|---|---|---|---|---|---|---|---|
| | Easy | Hard | Easy | Hard | Easy | Hard | Easy | Hard | Easy | Hard |
| *Adjust Bottle* | 81% | **75%** | 90% | 56% | 97% | 23% | 97% | 0% | **99%** | 3% |
| *Beat Block Hammer* | **77%** | **37%** | 43% | 21% | 56% | 3% | 42% | 0% | 72% | 8% |
| *Blocks Ranking RGB* | 3% | 0% | **19%** | **5%** | 1% | 0% | 0% | 0% | 3% | 0% |
| *Blocks Ranking Size* | 0% | 0% | **7%** | **1%** | 0% | 0% | 1% | 0% | 2% | 0% |
| *Click Alarmclock* | 61% | 12% | 63% | 11% | 32% | 4% | 61% | 5% | **77%** | **14%** |
| *Click Bell* | 80% | **9%** | 44% | 3% | 58% | 3% | 54% | 0% | **90%** | 0% |
| *Dump Bin Bigbin* | 64% | 32% | 83% | 24% | 68% | 1% | 49% | 0% | **85%** | **53%** |
| *Grab Roller* | 74% | 43% | 96% | **80%** | 94% | 25% | **98%** | 0% | 98% | 2% |
| *Handover Block* | 45% | **14%** | 45% | 8% | 42% | 0% | 10% | 0% | **70%** | 0% |
| *Handover Mic* | 90% | **31%** | 98% | 13% | 85% | 0% | 53% | 0% | **100%** | 3% |
| *Hanging Mug* | **23%** | **16%** | 11% | 3% | 7% | 0% | 8% | 0% | 17% | 1% |
| *Lift Pot* | 72% | 9% | 84% | **36%** | 88% | 0% | 39% | 0% | **97%** | 0% |
| *Move Can Pot* | 25% | 12% | 58% | **21%** | 22% | 4% | 39% | 0% | **70%** | 6% |
| *Move Pillbottle Pad* | 8% | 0% | 21% | **1%** | 0% | 0% | 1% | 0% | **41%** | 0% |
| *Move Playingcard Away* | 43% | 11% | 53% | **22%** | 36% | 0% | 47% | 0% | **68%** | 3% |
| *Move Stapler Pad* | 2% | 0% | 0% | **2%** | 0% | 0% | 1% | 0% | **12%** | 0% |
| *Open Laptop* | 59% | 32% | **85%** | **46%** | 56% | 0% | 49% | 0% | 82% | 7% |
| *Open Microwave* | 37% | 20% | 80% | **50%** | **86%** | 0% | 5% | 0% | 61% | 22% |
| *Pick Diverse Bottles* | 2% | 0% | 27% | **6%** | 7% | 0% | 6% | 0% | **52%** | 1% |
| *Pick Dual Bottles* | 42% | **13%** | 57% | 12% | 31% | 0% | 24% | 0% | **60%** | 1% |
| *Place A2B Left* | 3% | 1% | 31% | 1% | 1% | 0% | 2% | 0% | **46%** | **2%** |
| *Place A2B Right* | 1% | 1% | 27% | **6%** | 0% | 0% | 13% | 0% | **49%** | 0% |
| *Place Bread Basket* | 10% | 2% | 17% | **4%** | 6% | 0% | 14% | 0% | **26%** | 1% |
| *Place Bread Skillet* | 5% | **1%** | **23%** | **1%** | 7% | 0% | 11% | 0% | 19% | 0% |
| *Place Burger Fries* | 50% | **27%** | **80%** | 4% | 49% | 0% | 72% | 0% | 72% | 18% |
| *Place Can Basket* | 19% | **6%** | 41% | 5% | 1% | 0% | 18% | 0% | **67%** | 2% |
| *Place Cans Plasticbox* | 6% | **5%** | 34% | 2% | 16% | 0% | 40% | 0% | **48%** | 3% |
| *Place Container Plate* | 78% | 17% | **88%** | **45%** | 72% | 1% | 41% | 0% | 86% | 1% |
| *Place Dual Shoes* | 4% | **4%** | **15%** | 0% | 9% | 0% | 8% | 0% | 13% | 0% |
| *Place Empty Cup* | 56% | 7% | 37% | **11%** | 61% | 0% | 37% | 0% | **65%** | 1% |
| *Place Fan* | 12% | 2% | 20% | **10%** | 1% | 0% | 3% | 0% | **36%** | 1% |
| *Place Mouse Pad* | 1% | 0% | **7%** | **1%** | 0% | 0% | 0% | 0% | 4% | **1%** |
| *Place Object Basket* | 33% | **17%** | 16% | 2% | 15% | 0% | 15% | 0% | **65%** | 0% |
| *Place Object Scale* | 1% | **0%** | **10%** | **0%** | 0% | **0%** | 1% | **0%** | 15% | **0%** |
| *Place Object Stand* | 15% | 5% | 36% | **11%** | 1% | 0% | 22% | 0% | **60%** | 0% |
| *Place Phone Stand* | 15% | 6% | 35% | **7%** | 2% | 0% | 13% | 0% | **44%** | 2% |
| *Place Shoe* | 35% | **7%** | 28% | 6% | 5% | 0% | 23% | 0% | **58%** | 2% |
| *Press Stapler* | 41% | 24% | 62% | **29%** | 31% | 6% | 6% | 0% | **69%** | 3% |
| *Put Bottles Dustbin* | 21% | 4% | 54% | 13% | 27% | 1% | 22% | 0% | **60%** | **21%** |
| *Put Object Cabinet* | 33% | **18%** | 68% | **18%** | 15% | 0% | 42% | 0% | **72%** | 1% |
| *Rotate QRcode* | 50% | 5% | 68% | **15%** | 1% | 0% | 13% | 0% | **74%** | 1% |
| *Scan Object* | 4% | **1%** | 18% | **1%** | 2% | 0% | 9% | 0% | **31%** | **1%** |
| *Shake Bottle Horizontally* | 84% | **51%** | 99% | **51%** | 63% | 4% | 59% | 18% | **100%** | 25% |
| *Shake Bottle* | 74% | 45% | 97% | **60%** | 74% | 10% | 65% | 8% | **98%** | 19% |
| *Stack Blocks Three* | 2% | **0%** | **17%** | **0%** | 0% | **0%** | 0% | **0%** | 1% | **0%** |
| *Stack Blocks Two* | 21% | **2%** | **42%** | 1% | 25% | 0% | 7% | 0% | 24% | 0% |
| *Stack Bowls Three* | 51% | 17% | **66%** | **24%** | 48% | 0% | 63% | 0% | 57% | 5% |
| *Stack Bowls Two* | 76% | 30% | **91%** | **41%** | 82% | 0% | 61% | 0% | 83% | 6% |
| *Stamp Seal* | 1% | 0% | 3% | **4%** | 2% | 0% | 2% | 0% | **18%** | 0% |
| *Turn Switch* | 35% | 15% | 27% | **23%** | 5% | 2% | 36% | 1% | **46%** | 8% |
| *Average (%)* | 34.5 | 13.7 | 46.4 | **16.3** | 29.7 | 1.7 | 28.0 | 0.6 | **55.2** | 5.0 |

## A.20 Success Rates of Different Embodiments on RoboTwin 2.0 Tasks

Table 16 reports the success rates of five robot embodiments across the 50 RoboTwin 2.0 tasks, using the same set of expert programs for data generation.

Table 16: **Success Rates of Different Embodiments on RoboTwin 2.0 Tasks.**

| Task Name | RoboTwin1.0 | | | | | RoboTwin2.0 | | | | |
|---|---|---|---|---|---|---|---|---|---|---|
| | Aloha | ARX | Franka | Piper | UR5 | Aloha | ARX | Franka | Piper | UR5 |
| *Adjust Bottle* | 92% | 88% | 39% | 0% | 7% | 93% | 94% | 34% | 0% | 12% |
| *Beat Block Hammer* | 68% | 86% | 95% | 0% | 86% | 64% | 93% | 98% | 15% | 90% |
| *Blocks Ranking Rgb* | 92% | 98% | 96% | 0% | 82% | 96% | 97% | 99% | 13% | 53% |
| *Blocks Ranking Size* | 90% | 95% | 92% | 0% | 60% | 96% | 97% | 89% | 7% | 38% |
| *Click Alarmclock* | 89% | 99% | 100% | 0% | 95% | 92% | 99% | 100% | 0% | 95% |
| *Click Bell* | 100% | 100% | 100% | 9% | 100% | 100% | 100% | 100% | 91% | 100% |
| *Dump Bin Bigbin* | 85% | 98% | 90% | 0% | 82% | 84% | 100% | 84% | 9% | 80% |
| *Grab Roller* | 95% | 69% | 99% | 0% | 80% | 95% | 69% | 99% | 7% | 81% |
| *Handover Block* | 1% | 3% | 0% | 0% | 4% | 83% | 81% | 0% | 44% | 0% |
| *Handover Mic* | 62% | 80% | 92% | 28% | 0% | 87% | 98% | 84% | 65% | 14% |
| *Hanging Mug* | 68% | 76% | 5% | 0% | 12% | 63% | 73% | 11% | 0% | 11% |
| *Lift Pot* | 27% | 50% | 24% | 5% | 40% | 27% | 50% | 36% | 31% | 40% |
| *Move Can Pot* | 18% | 0% | 37% | 2% | 4% | 93% | 65% | 92% | 96% | 99% |
| *Move Pillbottle Pad* | 30% | 52% | 15% | 0% | 35% | 67% | 90% | 69% | 47% | 86% |
| *Move Playingcard Away* | 93% | 100% | 100% | 0% | 87% | 99% | 100% | 100% | 63% | 66% |
| *Move Stapler Pad* | 94% | 92% | 88% | 0% | 95% | 92% | 96% | 89% | 13% | 75% |
| *Open Laptop* | 76% | 91% | 78% | 14% | 55% | 82% | 92% | 77% | 23% | 51% |
| *Open Microwave* | 65% | 85% | 75% | 5% | 33% | 96% | 80% | 59% | 2% | 23% |
| *Pick Diverse Bottles* | 11% | 1% | 0% | 0% | 0% | 51% | 2% | 0% | 27% | 4% |
| *Pick Dual Bottles* | 8% | 3% | 0% | 0% | 0% | 92% | 6% | 0% | 81% | 7% |
| *Place A2B Left* | 65% | 75% | 70% | 0% | 72% | 80% | 88% | 64% | 29% | 76% |
| *Place A2B Right* | 70% | 68% | 68% | 0% | 69% | 81% | 82% | 64% | 31% | 66% |
| *Place Bread Basket* | 91% | 91% | 69% | 0% | 78% | 89% | 88% | 62% | 1% | 67% |
| *Place Bread Skillet* | 31% | 28% | 42% | 0% | 42% | 34% | 26% | 42% | 0% | 37% |
| *Place Can Basket* | 47% | 1% | 38% | 0% | 11% | 70% | 28% | 61% | 0% | 3% |
| *Place Cans Plasticbox* | 96% | 93% | 98% | 0% | 11% | 100% | 96% | 85% | 0% | 82% |
| *Place Container Plate* | 86% | 85% | 83% | 0% | 82% | 89% | 86% | 86% | 37% | 81% |
| *Place Dual Shoes* | 73% | 28% | 36% | 0% | 40% | 77% | 31% | 41% | 1% | 32% |
| *Place Empty Cup* | 92% | 100% | 100% | 0% | 100% | 92% | 100% | 100% | 4% | 100% |
| *Place Fan* | 93% | 96% | 75% | 0% | 85% | 95% | 93% | 83% | 0% | 65% |
| *Place Burger Fries* | 96% | 95% | 85% | 0% | 78% | 97% | 98% | 80% | 36% | 74% |
| *Place Mouse Pad* | 100% | 80% | 99% | 2% | 96% | 99% | 89% | 100% | 23% | 73% |
| *Place Object Basket* | 68% | 13% | 68% | 0% | 30% | 74% | 14% | 61% | 0% | 7% |
| *Place Object Scale* | 77% | 93% | 94% | 0% | 87% | 78% | 92% | 82% | 2% | 76% |
| *Place Object Stand* | 90% | 92% | 81% | 0% | 90% | 97% | 99% | 81% | 9% | 92% |
| *Place Phone Stand* | 66% | 78% | 52% | 22% | 44% | 66% | 78% | 45% | 53% | 49% |
| *Place Shoe* | 87% | 85% | 70% | 0% | 97% | 84% | 85% | 74% | 7% | 91% |
| *Press Stapler* | 87% | 96% | 99% | 0% | 77% | 98% | 96% | 100% | 59% | 72% |
| *Put Bottles Dustbin* | 0% | 0% | 0% | 0% | 0% | 71% | 1% | 0% | 56% | 0% |
| *Put Object Cabinet* | 13% | 56% | 43% | 0% | 0% | 14% | 24% | 55% | 0% | 0% |
| *Rotate Qrcode* | 78% | 83% | 98% | 0% | 81% | 75% | 74% | 94% | 0% | 67% |
| *Scan Object* | 8% | 13% | 21% | 0% | 8% | 4% | 45% | 26% | 0% | 19% |
| *Shake Bottle* | 62% | 95% | 82% | 1% | 98% | 89% | 94% | 85% | 74% | 97% |
| *Shake Bottle Horizontally* | 64% | 93% | 81% | 1% | 97% | 90% | 94% | 85% | 74% | 98% |
| *Stack Blocks Three* | 98% | 97% | 95% | 0% | 83% | 94% | 96% | 80% | 0% | 51% |
| *Stack Blocks Two* | 99% | 99% | 100% | 0% | 94% | 98% | 99% | 96% | 2% | 68% |
| *Stack Bowls Three* | 27% | 64% | 76% | 0% | 76% | 43% | 58% | 82% | 0% | 81% |
| *Stack Bowls Two* | 63% | 84% | 88% | 0% | 94% | 78% | 82% | 88% | 4% | 94% |
| *Stamp Seal* | 46% | 91% | 95% | 0% | 100% | 56% | 91% | 4% | 37% | 100% |
| *Turn Switch* | 27% | 3% | 51% | 28% | 10% | 74% | 3% | 36% | 81% | 10% |
| **Average** | **65.3%** | **68.8%** | **67.6%** | **2.3%** | **57.7%** | **78.8%** | **74.2%** | **67.2%** | **25.1%** | **57.1%** |
| **Difference** | / | / | / | / | / | **+13.5%** | **+5.4%** | **-0.4%** | **+22.8%** | **-0.6%** |

## A.21    50 ROBOTWIN 2.0 TASKS DESCRIPTIONS

We list detailed 50 RoboTwin 2.0 tasks descriptions in Tab. 17.

Table 17: Task descriptions.

| Task Name | Description |
|---|---|
| Adjust Bottle | Pick up the bottle on the table upright with the correct arm. |
| Beat Block Hammer | Grab the hammer and hit the block. |
| Blocks Ranking RGB | Order red, green, blue blocks from left to right. |
| Blocks Ranking Size | Order three blocks by size, largest to smallest, left to right. |
| Click Alarmclock | Click the center of the top button of the alarm clock. |
| Click Bell | Click the bell's top center. |
| Dump Bin Bigbin | Pour balls from the small bin into the big bin. |
| Grab Roller | Use both arms to grab the roller on the table. |
| Handover Block | Take the red block, hand it to the other arm, and place it on the pad. |
| Handover Mic | One arm passes the microphone to the other arm. |
| Hanging Mug | Left arm places mug; right arm picks it up and hangs it on the rack. |
| Lift Pot | Use the arms to lift the pot. |
| Move Can Pot | Pick up the can and move it next to the pot. |
| Move Playingcard Away | Move the playing card farther away from the table. |
| Move Stapler Pad | Move the stapler to a colored mat with the correct arm. |
| Open Laptop | Open the laptop with one arm. |
| Open Microwave | Open the microwave with one arm. |
| Pick Diverse Bottles | Pick up one bottle with each arm. |
| Pick Dual Bottles | Pick up one bottle with each arm. |
| Place A2B Left | Place object A to the left of object B. |
| Place A2B Right | Place object A to the right of object B. |
| Place Bread Basket | One bread: one arm to basket; two: both arms to basket. |
| Place Bread Skillet | Put the bread into the skillet with one arm. |
| Place Burger Fries | Use both arms to place burger and fries on the tray. |
| Place Can Basket | One arm puts can in basket; other arm lifts basket. |
| Place Cans Plasticbox | Use both arms to place cans into the plastic box. |
| Place Container Plate | Place the container on the plate. |
| Place Dual Shoes | Both arms place two shoes in box with tips facing left. |
| Place Empty Cup | Place the empty cup on the coaster with one arm. |
| Place Fan | Place the fan on the colored mat facing the robot. |
| Place Mouse Pad | Place the mouse on a colored mat. |
| Place Object Basket | One arm puts object in basket; other arm moves basket slightly away. |
| Place Object Scale | Place the object on the scale with one arm. |
| Place Object Stand | Place the object on the stand with the correct arm. |
| Place Phone Stand | Place the phone on the phone stand. |
| Place Shoe | Place the shoe from table onto the mat. |
| Press Stapler | Press the stapler with one arm. |
| Put Bottles Dustbin | Put bottles into the dustbin left of the table. |
| Put Object Cabinet | Open drawer with one arm; use other arm to put object inside. |
| Rotate QRcode | Rotate the QR board so the QR code faces the robot. |
| Scan Object | One arm holds scanner; other holds object; scan the object. |
| Shake Bottle Horizontally | Shake the bottle horizontally with the correct arm. |
| Shake Bottle | Shake the bottle with the correct arm. |
| Stack Blocks Three | Stack blue on green on red at the center. |
| Stack Blocks Two | Stack green on red at the center. |
| Stack Bowls Three | Stack three bowls. |
| Stack Bowls Two | Stack two bowls. |
| Stamp Seal | Stamp the specific color mat with the stamp. |
| Turn Switch | Click the switch with the arm. |
| Move Pillbottle Pad | Place the pill bottle onto the pad. |

## A.22 EXAMPLE: EXPERT CODE GENERATION PIPELINE FOR HANDOVER_BLOCK.

To make the pipeline in Fig. 3 more concrete, we use the `handover_block` task as a running example.

**Inputs and assumptions (first round).** In the first round, the inputs to the Code Agent include:

1. **Task description.** A natural language specification of the task, for example:

   "Use the left arm to pick up the block, move it to the handover position between the two arms, then use the right arm to grasp the block and place it at the target location."

2. **API list.** A fixed set of commonly used, high level, and strongly encapsulated APIs (listed in Appendix A.23), which hide low level motion planning details from the agent.

3. **API usage examples.** A small collection of curated examples that demonstrate how to use these APIs for typical scenarios and tasks, serving as in context guidance for the Code Agent.

4. **Object calibration and functional points.** Structured information about calibrated points and axes on the objects, including:

   • several grasp points on the block (for left arm and right arm grasps),
   • functional points on the block (for alignment and placement),
   • the functional point of the target placement location.

Given these inputs, the Code Agent produces executable expert control code that calls the high level APIs to complete the task.

**Outputs and iterative refinement.** We then execute the generated code in simulation to test data generation, as illustrated in Fig. 3 (10 test rollouts in our default setting). For each rollout, we log:

   • whether the rollout succeeds or fails,
   • the failure type (for example, `Left grasp failure`, `Right grasp failure`, `Incorrect target position`),
   • any runtime errors,
   • snapshot images of the scene after key steps.

If the data generation success rate is above a predefined threshold (50% in Fig. 3), we accept the code as a valid expert policy for data collection. If the success rate is below the threshold, we feed all interaction logs (failures, error messages, and snapshot descriptions) back to the Code Agent, which then iteratively refines the code to fix the identified issues. This loop continues until the generated code reaches the required success rate.

## A.23 ROBOTWIN 2.0 PROMPT

```
# You can directly use the actors provided in the actor_list.
# For example, if actor_list contains ["self.hammer", "self.block"],
# you can directly write:
object1 = self.hammer
object2 = self.block

# ------------------------------------------------------------
# Using ArmTag class to represent arms
# ------------------------------------------------------------
arm_tag = ArmTag("left") # Left arm
arm_tag = ArmTag("right") # Right arm

# Example of selecting an arm based on conditions:
arm_tag = ArmTag("left" if actor_position[0] < 0 else "right")

# ------------------------------------------------------------
```

```
# Functional points on actors
# -----------------------------------------------------------
# Each actor in the environment may have multiple functional points
# that are useful for different interactions.
# Functional points provide precise locations for interactions like
# grasping, placing, or aligning objects.

# To get a functional point from an actor:
functional_point_pose = actor.get_functional_point(point_id, "pose")
position = functional_point_pose.p # [x, y, z]
orientation = functional_point_pose.q # [qw, qx, qy, qz]

# -----------------------------------------------------------
# Stacking one object on top of another
# -----------------------------------------------------------
# Example: placing current_actor on top of last_actor using a
# functional point as target_pose.
target_pose = self.last_actor.get_functional_point(point_id, "pose")

self.move(
    self.place_actor(
        actor=self.current_actor, # The object to be placed
        target_pose=target_pose, # Pose acquired from last_actor
        arm_tag=arm_tag,
        functional_point_id=0, # Align functional point 0 (or as needed)
        pre_dis=0.1,
        dis=0.02,
        pre_dis_axis="fp", # Use functional point direction
    )
)

# -----------------------------------------------------------
# Actors of type "pose" in actor_list
# -----------------------------------------------------------
# For all actors in actor_list that are of type "pose", such as
# middle_pose or actor_target_pose, these are already Pose objects
# (or lists of Pose). You do NOT need to call .get_pose() again.
# You can pass them directly as target_pose.

# Example: place self.box at self.actor_pose (already a Pose)
self.move(
    self.place_actor(
        actor=self.box,
        target_pose=self.actor_pose, # already a Pose
        arm_tag=grasp_arm_tag,
        functional_point_id=0, # if the actor has functional points
        pre_dis=0,
        dis=0, # dis = 0 if is_open is False
        is_open=False, # gripper will not open after placing
        constrain="free", # "align" only if pose is constrained
        pre_dis_axis='fp', # use functional point direction
    )
)

# Note:
# For the target_actor, it is an actor, not a Pose, so you need to call
# get_pose() to get its pose, or call get_functional_point() to get
# a functional point.

# -----------------------------------------------------------
# Selecting an arm for grasping based on actor position
# -----------------------------------------------------------
# Get the actor's pose
actor_pose = self.actor.get_pose()
actor_position = actor_pose.p # [x, y, z]
```

```
# Select arm based on x-position
arm_tag = ArmTag("left" if actor_position[0] < 0 else "right")

# Grasp actor with selected arm
self.move(
    self.grasp_actor(
        actor=self.actor,
        arm_tag=arm_tag
    )
)

# ------------------------------------------------------------
# Basic grasping API examples
# ------------------------------------------------------------
# Grasp an actor with specified pre-grasp distance and grasp distance
self.move(
    self.grasp_actor(
        actor=self.actor,
        arm_tag=arm_tag, # ArmTag("left") or ArmTag("right")
        pre_grasp_dis=0.1,
        grasp_dis=0
    )
)

# ------------------------------------------------------------
# Grasp-and-lift example
# ------------------------------------------------------------
# Grasp the object
self.move(
    self.grasp_actor(
        actor=self.actor,
        arm_tag=arm_tag, # ArmTag("left") or ArmTag("right")
        pre_grasp_dis=0.1,
        grasp_dis=0
    )
)

# Lift the object up (always lift after grasping to avoid collision)
self.move(
    self.move_by_displacement(
        arm_tag=arm_tag,
        z=0.07, # Move 7cm upward
        move_axis='world' # Move in world coordinates
    )
)

# ------------------------------------------------------------
# Gripper control examples
# ------------------------------------------------------------
# Open gripper fully
self.move(
    self.open_gripper(
        arm_tag=arm_tag,
        pos=1.0 # fully open
    )
)

# Open gripper halfway
self.move(
    self.open_gripper(
        arm_tag=arm_tag,
        pos=0.5
    )
)
```

```
# Close gripper fully
self.move(
    self.close_gripper(
        arm_tag=arm_tag,
        pos=0.0 # fully close
    )
)

# Close gripper halfway
self.move(
    self.close_gripper(
        arm_tag=arm_tag,
        pos=0.5
    )
)

# ------------------------------------------------------------
# Placing objects at a target location
# ------------------------------------------------------------
# Place an object at a specific target pose
self.move(
    self.place_actor(
        actor=self.actor,
        arm_tag=arm_tag,
        target_pose=self.target_pose, # retrieved from the actor list
        functional_point_id=0, # if the actor has functional points
        pre_dis=0.1,
        dis=0.02, # dis = 0 if is_open is False
        is_open=True, # True to release object after placing
        pre_dis_axis='fp', # use functional point direction
    )
)

# Lift the gripper up after placing (only needed if is_open is True)
self.move(
    self.move_by_displacement(
        arm_tag=arm_tag,
        z=0.07, # Move 7cm upward
        move_axis='world'
    )
)

# ------------------------------------------------------------
# Placing with functional point alignment
# ------------------------------------------------------------
# Place the object by aligning functional point 0 with the target pose
self.move(
    self.place_actor(
        actor=self.actor,
        arm_tag=arm_tag,
        target_pose=target_pose,
        functional_point_id=0, # align this functional point
        pre_dis=0.1,
        dis=0.02,
        pre_dis_axis='fp' # use functional point direction
    )
)

# ------------------------------------------------------------
# Dual-arm coordination examples
# ------------------------------------------------------------
# Move both arms simultaneously to grasp objects
left_arm_tag = ArmTag("left")
right_arm_tag = ArmTag("right")
```

```
self.move(
    self.grasp_actor(actor=self.left_actor, arm_tag=left_arm_tag),
    self.grasp_actor(actor=self.right_actor, arm_tag=right_arm_tag)
)

# Lift both actors up after grasping
self.move(
    self.move_by_displacement(arm_tag=left_arm_tag, z=0.07),
    self.move_by_displacement(arm_tag=right_arm_tag, z=0.07)
)

# ------------------------------------------------------------
# Place left object while moving right arm back to origin
# ------------------------------------------------------------
move_arm_tag = ArmTag("left") # arm placing the object
back_arm_tag = ArmTag("right") # arm returning to origin

self.move(
    self.place_actor(
        actor=self.left_actor,
        arm_tag=move_arm_tag,
        target_pose=target_pose,
        pre_dis_axis="fp",
    ),
    self.back_to_origin(arm_tag=back_arm_tag)
)

# ------------------------------------------------------------
# Returning arms to their initial positions
# ------------------------------------------------------------
# Return a single arm to origin
self.move(self.back_to_origin(arm_tag=arm_tag))

# Return both arms to origin simultaneously
left_arm_tag = ArmTag("left")
right_arm_tag = ArmTag("right")

self.move(
    self.back_to_origin(arm_tag=left_arm_tag),
    self.back_to_origin(arm_tag=right_arm_tag)
)
```

Listing 4: LLM-Generated Examples for Actor, Functional Point, and Arm Control APIs

