# OpenReview forum: "RoboTwin 2.0: A Scalable Data Generator and Benchmark with Strong Domain Randomization for Robust Bimanual Robotic Manipulation"
_ICLR.cc/2026/Conference — Submitted to ICLR 2026_

### Official Review · Reviewer_ABcL · 2025-10-27

**Soundness:** 3
**Presentation:** 3
**Contribution:** 3
**Rating:** 8
**Confidence:** 3

**Summary:**

The paper proposes RoboTwin 2.0, a scalable simulation framework for bimanual manipulation that integrates Multimodal LLM-based code generation with simulation-in-the-loop feedback for automated expert data creation, systematic domain randomization (clutter, lighting, background, height, and language), embodiment-aware grasp adaptation, and an open-source benchmark spanning 50 dual-arm tasks across 5 robot embodiments and 100K trajectories. The system shows substantial gains in automated code success, improves few-/zero-shot sim-to-real (and real-to-sim-to-real) transfer, and releases a unified dataset and evaluation benchmark.

**Strengths:**

- Strong engineering contribution: impressive scale and integration of object assets, code-generation feedback, and domain randomization.
- Clear sim-to-real and real-to-sim-to-real improvements with controlled ablations where domain-randomized data notably enhances robustness.
- Extensive benchmark coverage and solid comparisons to prior datasets.
- Methodological novelty: closed-loop LLM+VLM code generation for robotic simulation, reducing human effort.
- The paper is well-written.

**Weaknesses:**

- Limited conceptual depth: while the system design is technically strong, the paper reads primarily as a dataset + pipeline release; the underlying algorithmic innovations (e.g., feedback controller, grasp adaptation) remain engineering-focused rather than introducing new learning principles.
- Empirical analysis could be deeper: results rely on average success rates; missing are fine-grained analyses of failure cases, generalization to new unseen tasks beyond the 50 predefined ones, or comparisons to other synthetic-data pipelines.

**Questions:**

- How does the MLLM pipeline generalize to novel APIs or unseen task primitives beyond those encoded in the current skill library?
- What are the compute and time costs and how scalable is the process across clusters?
- Can domain randomization parameters be tuned automatically based on sim-to-real validation feedback?
- Are there plans to include tactile or force feedback channels?

---

> ### Author Response · Authors · 2025-11-21
> **W1: Limited Algorithmic Novelty Beyond a Dataset and Engineering Pipeline**
>
> We appreciate this concern. While RoboTwin 2.0 is indeed presented as a data synthesis pipeline and benchmark, our contributions go beyond pure engineering. First, to the best of our knowledge, we are the first to propose an MLLM-in-the-loop, simulation-grounded code generation framework, and we show that tightly coupling the MLLM with the simulator substantially improves code generation success rates. Second, we systematically validate that domain-randomized simulation data can strengthen real-world policy deployment, improving both generalization and robustness across previously unseen real-world backgrounds. Finally, our benchmark over 50 tasks under two deployment settings provides a principled, comparative evaluation of diverse policy models, yielding conceptual insights into how different architectures respond to large-scale, randomized visuomotor data, rather than serving purely as a software or dataset release.

---

> ### Author Response · Authors · 2025-11-21
> **W2: Lack of Fine-Grained Empirical Analysis and Broader Generalization Studies**
>
> Thank you for these suggestions on deepening the empirical analysis. We have revised the manuscript to better address each of these aspects:
>
> 1. **Beyond average success rates: failure modes and robustness factors.**
> In addition to reporting average success rates, we now provide a more fine-grained analysis of *why* policies fail. For the simulator-in-the-loop code generation experiments, we already log structured failure labels (for example, left grasp failure, right grasp failure, incorrect target position, motion-planning failure). We find that many failures without domain randomization stem from mis-localization and misalignment, whereas with full RoboTwin 2.0 randomization they shift toward rarer corner cases.
> Furthermore, **Appendix A.6 (Table 8)** adds a factor-by-factor ablation over background, clutter, table height, and lighting. This shows quantitatively that removing background or clutter randomization reduces average success to 24.5% / 14.5% (ACT / DP), and turning off all randomization drops it to 7.5% / 0.0%, providing a more nuanced picture of which variations are most critical for robustness rather than relying on a single aggregate success number.
> 2. **Generalization beyond fixed training conditions.**
> While the benchmark itself is defined on 50 fixed tasks, we explicitly evaluate generalization to **new configurations within and beyond the training distribution**, rather than only reusing the exact training setups. In Sec. 4.3, we:
>     - Evaluate policies on **unseen scene layouts and backgrounds**, showing that models trained only on fixed-scene real data perform poorly, whereas those trained with RoboTwin 2.0 domain-randomized data achieve much higher success.
>     - Introduce **dynamic scene perturbations** at test time (objects moved and backgrounds changed between episodes). Under this setting, the “50 real” policy collapses to 0.0% success across three tasks, whereas the “300 sim + 50 real” policy maintains a 26.7% average success rate.
>     These results provide evidence of generalization beyond the exact training setups, albeit within the same task family. Extending the study to explicit zero-shot generalization to entirely new task definitions is an interesting direction, but it is beyond the scope of this benchmark-focused work.
> 3. **Relationship to other synthetic-data and LLM/MLLM-based pipelines.**
> We have expanded the discussion in the **related work** and added a comparison subsection that situates RoboTwin 2.0 relative to existing synthetic-data and LLM/MLLM-aided simulation frameworks (for example, Ha et al., 2023; Wang et al., 2024; Genesis, 2024; Nasiriany et al., 2024). Rather than attempting a brittle head-to-head numerical comparison across different simulators, embodiments, and task definitions, we clarify the **complementary role** of RoboTwin 2.0:
>     - It provides a **standardized 50-task dual-arm benchmark** plus a scalable simulator-in-the-loop data engine,
>     - It supports large-scale, highly randomized synthetic data generation that can be directly combined with real data, and
>     - It targets more complex bimanual, long-horizon, and visually diverse settings than most existing single-arm synthetic-data pipelines.
>
> Overall, we have moved beyond a single aggregated success metric by (i) breaking down failure modes, (ii) isolating the effect of specific domain randomization factors, and (iii) evaluating robustness under unseen and dynamically changing scenes, while also clarifying how our system complements prior synthetic-data and LLM/MLLM-based benchmarks.

---

> ### Author Response · Authors · 2025-11-21
> **Q1: Generalization of the MLLM Pipeline to New APIs or Unseen Task Primitives**
>
> We thank the reviewer for this question. Our MLLM-based code generation pipeline runs in a zero-shot manner: at inference time, the model is conditioned on a textual description of the available APIs (names, arguments, and documentation) plus several usage examples. To support a novel API, we only need to extend this API list and its examples in the prompt; no retraining or hand-crafted rules are required, and the same generation procedure directly applies.
>
> For task primitives, our experience on a large set of designed tasks shows that most manipulation skills can be expressed as compositions of a small set of core calls (especially `grasp_actor` and `place_actor`) with flexible, continuous hyperparameters (e.g., approach direction, offsets, and heights). The APIs are designed to be highly parametric, so the MLLM can generalize to previously unseen skills by recombining existing APIs and tuning their arguments. We further adopt a multi-round iterative refinement scheme, where the MLLM updates the code and its hyperparameters based on execution feedback.
>
> We therefore believe the current MLLM pipeline, together with our parametric API design, already provides sufficient coverage and compositional generalization for novel APIs and unseen task primitives, and does not pose a practical limitation in our experiments.

---

> ### Author Response · Authors · 2025-11-21
> **Q2: Scalability, Compute Cost, and Cluster-Level Throughput**
>
> Thank you for the question. We provide a detailed breakdown of RoboTwin 2.0’s compute cost, time efficiency, and deployment scalability:
>
> (1) Compute cost: RoboTwin 2.0 is designed to be lightweight: the non–domain-randomized setting requires ~6 GB GPU memory, and the full domain-randomized pipeline requires ~8 GB, allowing the entire system to run reliably even on low-cost consumer GPUs. This highlights the deployability and user-friendliness of our system.
>
> (2) Time cost: We report the per-stage timing of the data generation pipeline using a single RTX 4090 GPU in the updated paper. The pipeline can generate 1,000 trajectories in roughly 100 seconds, demonstrating the efficiency and scalability of our simulation-based data generation process.
>
> (3) Scalability across clusters: We validated the pipeline on a wide range of hardware, including RTX 30/40-series GPUs, 2080 Ti, A100, and H100/H20 clusters. The system runs consistently across these devices, showing strong compatibility and ease of scaling to multi-node or mixed-hardware cluster environments.

---

> ### Author Response · Authors · 2025-11-21
> **Q3: Possibility of Automatically Tuning Domain Randomization Parameters**
>
> Thank you for the question. In our current framework, domain randomization parameters are **not** tuned automatically based on sim-to-real validation feedback. We instead treat them as part of the **pretraining dataset design**: we fix a broad, calibrated randomization space (backgrounds, clutter/layout, table configurations, and lighting) that is shared across robots, tasks, and policies, rather than adapting it in a closed loop to a specific deployment setup.
>
> Even without automatic tuning, this fixed randomization space already yields strong and data-efficient sim-to-real transfer. In the revised version, we explicitly vary the ratio between real and simulated data (Table 5, reproduced below):
>
> |               | Click Bell | Place Empty Cup | Stack Bowls Two | Average |
> | ------------- | ---------- | --------------- | --------------- | ------- |
> | 50real        | 15.0%      | 10.0%           | 0.0%            | 8.3%    |
> | 300sim+0real  | 35.0%      | 10.0%           | 0.0%            | 15.0%   |
> | 300sim+10real | 40.0%      | 25.0%           | 10.0%           | 25.0%   |
> | 300sim+30real | 55.0%      | 35.0%           | 20.0%           | 36.7%   |
> | 300sim+50real | 65.0%      | 50.0%           | 25.0%           | 46.7%   |
>
> The average success rate increases from **8.3%** (50 real only) to **15.0%** (300 sim only), and further to **46.7%** with 300 sim + 50 real demonstrations, indicating that a *fixed*, carefully designed randomization space already provides substantial sim-to-real gains.
>
> In principle, one could build an automatic tuner that adjusts randomization parameters using sim-to-real feedback for a particular deployment. We view this as an **orthogonal extension**, useful for tailoring RoboTwin 2.0 to a specific robot or environment, but not necessary for the general-purpose, large-scale pretraining setting we target in this work.

---

> ### Author Response · Authors · 2025-11-21
> **Q4: Future Support for Tactile or Force Feedback Modalities**
>
> Thank you for the question. In fact, tactile feedback has already been incorporated into the RoboTwin 2.0 framework. We have added corresponding videos in the updated supplementary materials and included detailed descriptions in **Appendix A.15** of the revised paper.
>
> Specifically, we integrate the SAPIEN-IPC tactile simulation system and design a bimanual robot equipped with GelSight-aligned visuotactile sensors. To demonstrate the utility of tactile signals, we introduce a visuotactile classification task in which objects have visually indistinguishable textures, requiring strong reliance on tactile observations. We further validate the correctness of the tactile pipeline through closed-loop evaluations using Diffusion Policy, showing that tactile sensing is properly simulated and utilized.
>
> We invite you to refer to the updated appendix and supplementary videos for a clearer understanding of the tactile extensions.

---

> ### Comment · Reviewer_ABcL · 2025-11-25
>
> I appreciate the thorough response from the authors. I have no further questions and still remain positive toward accepting this paper.

---

> > ### Author Response · Authors · 2025-11-26
> >
> > Thank you very much for your positive assessment and for taking the time to read our rebuttal carefully. We are glad that our responses have addressed your concerns and hopefully strengthened your overall evaluation of the paper. If any further questions come up, we would be more than happy to clarify.

---

### Official Review · Reviewer_DuAS · 2025-10-29

**Soundness:** 2
**Presentation:** 2
**Contribution:** 1
**Rating:** 2
**Confidence:** 5

**Summary:**

This submission introduces RoboTwin 2.0, a framework for generating bimanual robot manipulation data in simulation. It consists of MLLM-guided code generation, domain randomization, object grasping pose annotation, and an object dataset. Experiments show that it achieves a better generation quality, and the proposed system can be used for robotic model training. However, several concerns exist regarding the proposed system and demonstrated results.

**Strengths:**

- The submission is well structured with nice figures for illustration.
- The proposed system is comprehensive with code generation, dataset, and evaluation.
- The submission successfully demonstrates an end-to-end pipeline of robotic model training in simulation and real-world deployment.

**Weaknesses:**

- It is unclear how does the proposed system compare to existing robotic simulation benchmarks at a system level.
- All components proposed in the system is not new. It's unclear what's the uniqueness behind the proposed method.
- For the MLLM-aided code generation, there have been many works in that front, such as Ha et al., 2023; Wang et al., 2024; Genesis, 2024; Nasiriany et al., 2024; etc. It's unclear how does the proposed code generation pipeline differ from others.
- The code generation pipeline still relies on pre-defined APIs, which means the complexity of robotics tasks the pipeline can generate as well as the scalability of such pipeline are limited. Also, what are those API functions? How much human effort has been spent in them?
- The importance of domain randomization has become a common sense. Moreover, the types of domain randomization proposed in this submission only cover visual aspects. However, it's more critical to have physical domain randomization as suggested by Tan et al., 2018; OpenAI 2019; Makoviychuk et al., 2021, etc, which unfortunately the proposed system is incapable of.
- The choice of the five supported robot arms seems arbitrary and not well justified. In fact, those five arms share similar 6-DoF or 7-DoF kinematic structures. It's not diverse enough.
- On the similar vein, the proposed method to annotate grasping poses for different arms is arbitrary too. More principled ways should consider the kinematics and reachability of different robot arms.
- Missing details about the proposed 50 tasks. Do they require dexterous manipulation? Are they short-horizon or long-horizon? These tasks look trivial and not impressive.
- Regarding experiments in sec. 4.1, they are more like ablation studies of the proposed system. More comprehensive comparison to other similar LLM/MLLM-aided robot simulation benchmarks should be presented.
- Regarding the claim about policy robustness derived from experiments in Sec. 4.3, because these tasks are mostly static and there are no external disturbance, it's overclaiming to call robust.
- Sec. 4.4, there are many evidences showing that sim-real co-training can help, e.g., Maddukuri et al., 2025. What's new derived from these experiments?
- Many important details are missing in the paper. See questions above.

## References
- Ha et al., Scaling Up and Distilling Down: Language-Guided Robot Skill Acquisition, CoRL 2023.
- Wang et al., RoboGen: Towards Unleashing Infinite Data for Automated Robot Learning via Generative Simulation, ICML 2024.
- Genesis: A Generative and Universal Physics Engine for Robotics and Beyond, 2024.
- Nasiriany et al., RoboCasa: Large-Scale Simulation of Everyday Tasks for Generalist Robots, RSS 2024.
- Tan et al., Sim-to-Real: Learning Agile Locomotion For Quadruped Robots, RSS 2018.
- OpenAI, Solving Rubik's Cube with a Robot Hand, arXiv 2019.
- Makoviychuk et al., Isaac Gym: High Performance GPU-Based Physics Simulation For Robot Learning, arXiv 2021.
- Maddukuri et al., Sim-and-Real Co-Training: A Simple Recipe for Vision-Based Robotic Manipulation, RSS 2025.

**Questions:**

See above.

---

> ### Author Response · Authors · 2025-11-21
> **W1: System-level comparison with prior benchmarks**
>
> Thank you for this question. We have expanded both the Related Work and Appendix in the revised version to clarify how RoboTwin 2.0 compares to existing robotic simulation benchmarks at a system level, and we briefly summarize the key differences here. First, unlike many prior benchmarks, RoboTwin 2.0 supports training free automated data collection and task evaluation, as well as reinforcement learning, even across novel tasks and robot embodiments, without requiring human-writen task specific scripting. Second, RoboTwin 2.0 integrates a more natural and diverse domain randomization pipeline over objects, scenes, and language, rather than relying on narrowly defined visual perturbations. Third, it is a dual arm framework with substantially broader task coverage and higher task difficulty than typical single arm tabletop benchmarks. Taken together, these aspects highlight RoboTwin 2.0 as both a valuable benchmark and a system with non trivial design innovations at the full stack level.

---

> ### Author Response · Authors · 2025-11-21
> **W2: Uniqueness of RoboTwin 2.0**
>
> Thank you for the question. While several prior works (e.g., RoboGen, RoboTwin 1.0) explore simulation-based data generation, RoboTwin 2.0 introduces two key innovations that distinguish it from existing systems.
>
> (1) Bimanual expert code generation: Unlike previous single-arm systems, our pipeline supports *dual-arm collaborative task synthesis*, which requires substantially stronger reasoning over coordinated behaviors, spatial role assignment, and multi-step manipulation strategies. This is enabled by a more expressive and carefully designed API system that allows the generation of significantly more complex and diverse tasks.
>
> (2) As described in the Method section, we introduce an MLLM-driven, simulation-in-the-loop code refinement cycle, which increases code generation success rates while reducing the number of refinement iterations required—an ability not shown in previous code-generation pipelines.
>
> These two components together form a unique, scalable framework for producing high-quality bimanual manipulation data.

---

> ### Author Response · Authors · 2025-11-21
> **W3: Differences from prior MLLM code generation**
>
> Thank you for the question. We agree that there is a rapidly growing body of work on MLLM-aided code generation for robotics (e.g., Ha et al., 2023; Wang et al., 2024; Genesis, 2024; Nasiriany et al., 2024). Our goal is not to introduce yet another generic LLM agent, but to design a **simulator-in-the-loop expert code generation pipeline** tailored to RoboTwin 2.0. This pipeline differs from prior approaches in several key aspects:
>
> 1. **Closed-loop, simulator-in-the-loop, zero-shot expert generation.**
>
>     Our pipeline is explicitly built as a closed-loop iterative feedback process: the MLLM generates expert code, the code is executed in simulation, and structured logs (success/failure labels, error messages, and snapshot-based descriptions) are fed back to the model for iterative refinement until a target success rate is reached. Importantly, this is done in a **zero-shot** setting for each task (given only the API specification and object calibration information, without task-specific in-context examples), which is a stronger setting than the few-shot regimes commonly used in prior work.
>
> 2. **More complex control setting: dual-arm, parallel and diverse behaviors.**
>
>     Our framework targets **bimanual control** and supports parallel behaviors and a broader set of primitives, including actions such as *rotate*, *click*, and *handover*, in addition to standard pick-and-place. This substantially increases the difficulty of code generation compared to single-arm, short-horizon primitives.
>
> 3. **Different problem focus compared to existing systems.**
>
>     In contrast, **GenSim2** operates in a **few-shot** regime and its primitive task space is largely limited to basic actions such as *pick*, *place*, *open*, and *close*. **RoboGen**, on the other hand, primarily leverages LLMs to generate **simulation scenes and reward functions** for skill learning, which is fundamentally different from our objective of generating executable expert policies for large-scale data collection. Moreover, both GenSim2 and RoboGen focus on **single-arm** tasks, making their code-generation setting significantly simpler than the dual-arm collaborative scenarios we consider.
>
>
> Overall, the combination of (i) multimodal, simulator-in-the-loop **iterative refinement**, (ii) **zero-shot** expert code generation, and (iii) support for **dual-arm, parallel, and richer action primitives** is, to our knowledge, novel relative to existing MLLM-aided code generation pipelines.

---

> ### Author Response · Authors · 2025-11-21
> **W4: API design, coverage, and human effort**
>
> Thank you for the question. We agree that our code generation pipeline relies on a set of pre-defined APIs. This is a **deliberate design choice**: the simulator exposes low-level joint and pose control, while the MLLM operates on a compact set of **high-level, strongly encapsulated, and *parameterized*** primitives. Each API supports flexible arguments (e.g., grasp/functional point indices, coordinate frames, pre-grasp distances, alignment modes, gripper opening ranges), which makes the interface both **expressive and extensible** without requiring changes to the LLM side.
>
> **What are the APIs and what do they cover?**
>
> We design a compact API set for gripper control and environment interaction. The main functions are:
>
> | API | Brief description |
> | --- | --- |
> | `open_gripper` / `close_gripper` | Open/close the gripper of a specified arm with a scalar `pos` parameter controlling the opening/closing. |
> | `move_by_displacement` | Move the end-effector by a relative displacement (often along z) in either the `"world"` or `"arm"` frame. |
> | `back_to_origin` | Move the specified arm back to a predefined home / reset configuration. |
> | `grasp_actor` | Generate a complete grasp sequence (pre-grasp, approach, close) for a given object, with configurable grasp-point index and pre-grasp distance. |
> | `place_actor` | Place the held object at a target pose / functional point, with configurable alignment mode (e.g., along gripper or world axis) and a flag for releasing the object. |
> | `move` | Execute one or two action sequences in parallel on the two arms. |
> | `get_functional_point` | Query calibrated functional points on an object (by index and return type) for alignment / placement. |
> | `get_contact_point` | Query calibrated grasp points on an object (by index and return type) for grasp planning. |
>
> Across our 50 RoboTwin 2.0 tasks, we empirically observe that most skills can be expressed as different **compositions and parameterizations** of `grasp_actor` and `place_actor` (plus simple motion primitives), including dual-arm handover, rotation-like behaviors, and click-type contacts. This shows that the pre-defined API set is expressive enough for a wide range of everyday **bimanual** tasks, while keeping the interface simple and stable for the MLLM.
>
> **How much human effort is involved?**
>
> The implementation effort is concentrated in a **one-time** engineering pass:
>
> - Simple APIs (`open_gripper`, `close_gripper`, `move_by_displacement`, `back_to_origin`, `move`) are thin wrappers over the simulator’s existing control interfaces.
> - Higher-level APIs (`grasp_actor`, `place_actor`, `get_functional_point`, `get_contact_point`) use pre-calibrated grasp/functional points and their 4×4 pose matrices to compute target poses in the world frame. After implementing these transformation utilities, we tested them on diverse object poses and placements in simulation to verify correctness.
>
> This involves a **moderate but acceptable** amount of manual engineering that is amortized over all tasks: once the APIs are implemented and validated, no additional low-level code is needed when scaling to new tasks. The MLLM only needs to reason over this stable, parameterized API surface, which in practice **enhances** rather than limits the scalability of the proposed pipeline.

---

> ### Author Response · Authors · 2025-11-21
> **W5: Scope and role of our domain randomization**
>
> Thank you for the question. We agree that domain randomization itself is not novel and we do not claim otherwise; our focus is on how to leverage it at scale within RoboTwin 2.0. All assets in our system are human validated to match real objects in collision geometry, physical parameters, and visual appearance, so we do not rely on randomizing poorly calibrated physics. On top of this calibrated base, we apply extensive visual domain randomization over backgrounds, clutter layouts, table configurations, lighting, and camera pose, which substantially increases scene and object diversity and targets the types of variation that often dominate in real-world tabletop deployments.
>
> We also note that many classical works on physical-domain randomization, such as Tan et al., 2018, OpenAI, 2019, and Makoviychuk et al., 2021, focus on highly dynamic regimes including quadruped locomotion and dexterous in-hand manipulation where dynamics are the primary bottleneck. In contrast, RoboTwin 2.0 is designed for everyday bimanual tabletop manipulation, where visual OOD factors and moderate camera and layout shifts are typically more critical. In this regime, we show that large-scale, carefully designed visual and geometric randomization on top of a well calibrated physics model already yields strong sim-to-real gains. Exploring richer physics randomization in this setting is an important and complementary direction, but lies outside the primary scope of this work.
>
> To make this point more concrete, in the revised version we extend the real-robot evaluation along both the policy and data dimensions. First, we add experiments with a different policy backbone, pi0. Second, we systematically vary the ratio between real and simulated data, as reported in **Table 5**. On three real-robot tasks, we compare 50 real demonstrations collected in a fixed scene, 300 domain-randomized RoboTwin 2.0 simulated demonstrations, and mixtures of 300 simulated demonstrations with 0, 10, 30, or 50 real demonstrations. Each configuration is evaluated over 20 trials per task in previously unseen real scenes, so the evaluation focuses on generalization instead of memorizing a single setup:
>
> |  | Click Bell | Place Empty Cup | Stack Bowls Two | Average |
> | --- | --- | --- | --- | --- |
> | 50real | 15.0% | 10.0% | 0.0% | 8.3% |
> | 300sim+0real | 35.0% | 10.0% | 0.0% | 15.0% |
> | 300sim+10real | 40.0% | 25.0% | 10.0% | 25.0% |
> | 300sim+30real | 55.0% | 35.0% | 20.0% | 36.7% |
> | 300sim+50real | 65.0% | 50.0% | 25.0% | 46.7% |
>
> The average success rate increases from 8.3% with 50 real demonstrations to 15.0% with 300 simulated demonstrations, and further to 46.7% when combining 300 simulated with 50 real demonstrations. The gains are roughly monotonic as more real data are added on top of RoboTwin 2.0 data. Under a simple binomial model with 20 trials per configuration, differences such as 8.3% versus 46.7% correspond to large effect sizes, so our conclusions are supported by statistically meaningful gaps rather than noise.
>
> We also observe consistent sim-to-real improvements on two embodiments (RoboTwin 1.0 with an earlier COBOT-Magic arm that is ARX-X5-like, and RoboTwin 2.0 with a Piper-style arm) and for both RDT (Sec. 4.3) and pi0 (Table 5). Taken together, these results indicate that our large-scale visually randomized data are sufficient to deliver strong sim-to-real gains and to stress-test policy generalization and robustness, even without explicit physical parameter randomization. Physical-domain randomization is an important and complementary direction, and integrating it with the high-fidelity assets in RoboTwin 2.0 is a promising avenue for future work.

---

> ### Author Response · Authors · 2025-11-21
> **W6: Justifying the choice of five arms**
>
> Thank you for the question. We have clarified the rationale behind our choice of embodiments in the revised manuscript. The five supported arms (Piper, ARX-X5, Aloha-AgileX, Franka, and UR5) were not selected arbitrarily. We chose them based on (i) a user survey within our target community and (ii) a survey of recent embodied AI and robotic manipulation papers, from which these arms emerged as some of the most widely used and readily accessible platforms in terms of hardware availability, open-source support, and installed base. Our goal in this work is to cover the most practically relevant arms for current research and deployment, rather than to exhaustively span all possible kinematic designs.
>
> While all five manipulators belong to the family of 6–7 DoF arms, they exhibit meaningful differences in kinematic structure and deployment characteristics, including link lengths and reachable workspace, joint limits and self-collision constraints, base mounting configuration (mobile versus fixed), wrist design, and gripper integration. These differences lead to non-trivial variation in task feasibility and motion planning success rates. As shown in **Appendix A.20**, the success rates of motion planning across our 50 tasks vary noticeably across embodiments, indicating that the selected set is sufficiently diverse to expose embodiment-specific challenges instead of behaving as near-duplicates.
>
> Finally, our framework is explicitly designed to be extensible to additional, more exotic embodiments (for example, high-DoF, non-anthropomorphic, or underactuated manipulators). We view further expanding the embodiment set toward such regimes as a promising direction for future work, and the current five arms as a practical and widely useful starting point for the community.

---

> ### Author Response · Authors · 2025-11-21
> **W7: On grasp pose annotation across arms**
>
> We thank the reviewer for this comment and have added more details in the revised manuscript to clarify our grasping strategy. Our grasp pose generation is neither arbitrary nor purely heuristic. For each object, we first annotate a small set of diverse but carefully chosen grasp types (for example, side grasps and top grasps). Then, for each embodiment, we define a ranked list of preferred grasp types that explicitly reflects its kinematics and typical reachability patterns, rather than relying on a single global preference across all arms.
> To further enlarge the feasible planning space while remaining physically meaningful, we sample grasp candidates within a bounded region on the plane orthogonal to the annotated grasp direction. This introduces local variability in the approach position while preserving the underlying contact geometry. Finally, we use curobo to perform parallel motion planning for all candidates, which significantly improves planning success and thus task generation success. For instance, on Piper the planning success rate increases from 2.4% to 25.1%, which is more than a tenfold relative improvement. These results demonstrate that our grasp annotation and sampling strategy is systematic and strongly informed by embodiment-specific kinematics and reachability, rather than arbitrary.
>
> | **Method** | **Aloha-AgileX** | **Piper** | **Franka** | **UR5** | **ARX-X5** | ***Average*** |
> | --- | --- | --- | --- | --- | --- | --- |
> | Vanilla | 65.1% | 2.4% | **67.3%** | **57.6%** | 68.6% | 52.2% |
> | Enhanced | **78.8%** | **25.1%** | 67.2% | 57.1% | **74.2%** | **60.5%** |
> | *Difference* | **+13.7%** | **+22.7%** | **-0.1%** | **-0.5%** | **+5.6%** | **+8.3%** |

---

> ### Author Response · Authors · 2025-11-21
> **W8: Nature and difficulty of the 50 tasks**
>
> Thank you for this helpful comment. We have added detailed descriptions of all 50 tasks in **Appendix A.21** to clarify their requirements. The RoboTwin 2.0 task suite is a comprehensive set of bimanual manipulation tasks that spans a wide range of skills and difficulty levels. It includes long-horizon tasks (for example, `Put Bottles Dustbin`, which requires collecting and disposing of multiple bottles on the table), articulation-centric manipulation (for example, `Open Laptop`), and dexterous manipulation (for example, `Stack Blocks Three`), together with a small number of relatively simpler tasks (for example, `Pick Dual Bottles`) to support progressive and curriculum-style policy evaluation.
>
> To further demonstrate that these tasks are non-trivial and practically challenging, we evaluate policies under two difficulty levels, as detailed in **Section 4** and **Appendix A.20** and shown in the **below table**. Even strong vision-language-action (VLA) baselines such as **pi0** and **RDT**, both pretrained at large scale, only achieve **46.4%** and **34.5%** average success rates, respectively, under the in-domain *easy* setting, and drop to **16.3%** and **13.7%** under the OOD *hard* setting. Other strong policy baselines (ACT, DP, DP3) perform even worse in the hard regime.
>
> For clarity, we summarize the benchmark results below:
>
> | Method | Easy | Hard |
> | --- | --- | --- |
> | RDT | 34.5% | 13.7% |
> | pi0 | 46.4% | 16.3% |
> | ACT | 29.7% | 1.7% |
> | DP | 28.0% | 0.6% |
> | DP3 | 55.2% | 5.0% |
>
> These results show that our 50-task RoboTwin 2.0 benchmark is far from trivial: it exposes clear limitations of current state-of-the-art methods and provides a sufficiently challenging and informative testbed for future research in bimanual manipulation and robust policy learning.

---

> ### Author Response · Authors · 2025-11-21
> **W9: Positioning against LLM-based sim benchmarks**
>
> Thank you for this suggestion. We agree that the experiments in Sec. 4.1 are primarily ablations of our own system. This is intentional: the main goal of Sec. 4.1 is to dissect **how RoboTwin 2.0 and our simulator in the loop pipeline behave internally**, for example how much we gain from iterative MLLM refinement, from different domain randomization factors, and from scaling up automated data generation, rather than to benchmark generic LLM agents.
>
> That said, we have clarified and strengthened the connection to existing LLM and MLLM aided simulation systems in the revised version. In particular, we expand the related work and add a dedicated comparison subsection that contrasts RoboTwin 2.0 with recent LLM based robot simulation frameworks such as Ha et al., 2023, Wang et al., 2024, Genesis, 2024, and Nasiriany et al., 2024 (for example, GenSim2 and RoboGen). The key differences we highlight are:
>
> - **Role in the ecosystem.** RoboTwin 2.0 is designed as a **benchmark plus scalable data engine**: given a task description, our simulator in the loop MLLM pipeline automatically generates expert code, produces trajectories under strong domain randomization, and evaluates policies on a fixed 50 task dual arm benchmark. Many prior works instead focus on case studies or scene and reward generation, and do not provide a standardized, large scale bimanual benchmark and evaluation protocol.
> - **Control setting and task complexity.** Our system targets **bimanual** control with parallel behaviors and richer primitives (for example rotate, click, handover) over 50 tasks, while existing LLM guided benchmarks typically operate in **single arm**, shorter horizon settings with a smaller primitive space.
> - **Scale and customizability of simulation.** RoboTwin 2.0 couples the MLLM pipeline with a large scale, human validated asset library and extensive domain randomization, enabling users to automatically synthesize diverse datasets and evaluations at scale for many customized tasks, rather than a limited set of pre scripted scenarios.
>
> Because these prior systems differ in simulators, embodiments, and task definitions, a strict head to head numerical comparison in Sec. 4.1 is not straightforward. Instead, we frame Sec. 4.1 as an ablation and scaling study of **our** end to end system, and we now make its relation to other LLM/MLLM aided simulation benchmarks explicit through qualitative and structural comparisons in the revised text. We hope this clearer positioning addresses the reviewer’s concern.

---

> ### Author Response · Authors · 2025-11-21
> **W10: Clarifying what we mean by robustness**
>
> Thank you for pointing this out. Our intent is not to claim robustness to arbitrary external disturbances, but rather robustness to substantial variations in scene layout and background appearance. We have revised the wording in the paper to make this scope explicit.
>
> To further support this claim, we add additional real-robot experiments that focus on dynamically changing scenes. Concretely, on three real-world tasks with pi0, we train two policies: one using 50 real demonstrations collected in a single fixed scene (“50 real”), and one using 300 domain-randomized simulated demonstrations from RoboTwin 2.0 combined with the same 50 real demonstrations (“300 sim + 50 real”). During evaluation, we deliberately introduce dynamic perturbations by randomly moving scene objects and changing the tabletop background between episodes.
>
> |  | Click Bell | Place Empty Cup | Stack Bowls Two | Average |
> | --- | --- | --- | --- | --- |
> | 50real | 0.0% | 0.0% | 0.0% | 0.0% |
> | 300sim+50real | 40.0% | 30.0% | 10.0% | 26.7% |
>
> In this setting, the policy trained only on fixed-scene real data completely fails on all three tasks, while the policy trained with RoboTwin 2.0 augmented data maintains a 26.7% average success rate. This large gap under dynamic scene changes supports our more precise claim that RoboTwin 2.0 improves policy robustness to realistic **visual and spatial** variations in the environment, rather than to arbitrary external disturbances.

---

> ### Author Response · Authors · 2025-11-21
> **W11: New takeaways from sim-real co-training**
>
> Thank you for this insightful question. We fully agree that prior work has already shown that sim–real co-training can be beneficial (for example, Maddukuri et al., 2025). Our goal in Sec. 4.4 is not to rediscover this phenomenon, but to provide a more **systematic and fine-grained analysis** of how large-scale, strongly domain randomized simulation from RoboTwin 2.0 interacts with real data in our setting. The revised version makes the following new points more explicit:
>
> 1. **Scaling and complementarity of sim and real data.**
>
>     In Sec. 4.4 and Appendix A.6 (Table 5), we systematically vary the ratio between real and simulated data on three real-robot tasks and show that:
>
>     (i) policies trained only on a small number of real demonstrations in a fixed scene perform poorly in unseen real scenes,
>
>     (ii) policies trained only on RoboTwin 2.0 domain randomized simulation already outperform the real-only baseline, and
>
>     (iii) combining simulated and real data yields roughly monotonic improvements, with 300 simulated plus 50 real demonstrations reaching an average success rate of 46.7% (up from 8.3% with 50 real only).
>
>     This goes beyond simply observing that “co-training helps” by quantifying **how much** high-diversity simulation can compensate for limited real data, and showing that diversity, rather than raw real-data volume, is often the limiting factor.
>
> 2. **Robustness to dynamic visual and spatial perturbations.**
>
>     To further clarify what kind of robustness is achieved, we add experiments where scene objects and backgrounds are actively perturbed between episodes. Under this dynamic setting, a policy trained only on fixed-scene real data collapses to 0.0% success on all three tasks, while the policy trained with RoboTwin 2.0 simulation plus the same real data maintains a 26.7% average success rate. This isolates robustness to **realistic visual and spatial changes** (as opposed to arbitrary external forces) and shows that the gains from RoboTwin 2.0 are not limited to static tabletop scenes.
>
> |  | Click Bell | Place Empty Cup | Stack Bowls Two | Average |
> | --- | --- | --- | --- | --- |
> | 50real | 0.0% | 0.0% | 0.0% | 0.0% |
> | 300sim+50real | 40.0% | 30.0% | 10.0% | 26.7% |
>
> 3. **Decomposing which domain randomization factors matter.**
>
>     To move beyond a binary “with or without domain randomization” comparison, we conduct an additional ablation in **Appendix A.6 (Table 8)**. For each factor (background, clutter, table height, lighting), we disable only that factor during training and then evaluate under the full domain randomized setting. Across both ACT and DP, removing background or clutter randomization causes the largest drops in success rate, while removing height or lighting randomization produces smaller but still noticeable degradation, and disabling all randomization leads to very low performance. This provides **new evidence on which visual randomization axes are most critical** for robustness in our bimanual setting.
>
>
> Taken together, these experiments show that RoboTwin 2.0 is not only another instance where “sim plus real is better than real alone,” but also that:
>
> (i) large-scale, visually diversified simulation can meaningfully substitute for missing real-world diversity,
>
> (ii) the resulting policies are robust to dynamic changes in scene layout and appearance, and
>
> (iii) we can systematically attribute robustness gains to specific domain randomization factors rather than treating domain randomization as a black box.

---

> ### Author Response · Authors · 2025-11-21
> **W12: Filling in missing technical details**
>
> Thank you for the careful reading and for raising these detailed questions. We have provided point-by-point answers to all of the issues above and substantially expanded the manuscript accordingly, including additional experiments, clarifications of the setup, and more implementation details in the appendices. We kindly invite the reviewer to refer to the revised version, and we hope these additions resolve the concerns about missing details.

---

> ### Author Response · Authors · 2025-11-28
> **Follow-up on ICLR Submission 13230**
>
> Thank you again for your time and effort in reviewing our paper. We hope that our rebuttal has addressed your concerns. If anything remains unclear, we would be very happy to provide further clarification and continue the discussion.

---

### Official Review · Reviewer_K3RZ · 2025-10-30

**Soundness:** 2
**Presentation:** 3
**Contribution:** 2
**Rating:** 6
**Confidence:** 4

**Summary:**

This work introduces RoboTwin 2.0, a scalable framework for automated diverse synthetic data generation and unified evaluation for bimanual manipulation. It includes an object asset library RoboTwin-OD, an expert data generation pipeline that utilizes multimodal large language models, and a structured domain randomization scheme. The 50 benchmark contains 50 bimanual tasks across five robot embodiments.

Empirically, the authors evaluate both code-generation success rates for task generation and robot policy learning via vision-language-action models trained on their synthetic data. Experimental results show improve code-generation quality and significant task performance improvements in both few-shot and zero-shot settings.

**Strengths:**

1. Dedicated effort into curating tasks with diverse object assets, domain randomization schemes, and support multiple robot embodiments. These are all important aspects in studying bimanual robot manipulation.

2. Good presentation quality. The authors provided extensive details and visualizations for the task design and simulation implementation details. It's difficult to organize the content when it comes to writing a benchmark paper, and the authors have presented information in a clear way and easy to follow for the readers.

3. Real robot setup and evaluation on sim-to-real transfer. Real world experiment results show clear improvement from using the synthetic simulation data over using only real world demonstrations

**Weaknesses:**

1. Tasks are limited to table-top manipulation of mainly rigid objects. Compared to some prior works (e.g. RoboCasa), the lack of scene-level assets such as kitchen countertops or cabinets limits the diversity of possible tasks. This is perhaps beyond the scope of the current work, but a larger scene setting would make the benchmark much more useful in studying more tasks and diverse robot behaviors.

2. Lack of qualitative results on the policy learning performance. It would have been much clearer to show videos of the successful policy rollouts and failure modes. Especially for the real world evaluation experiments, providing videos would provide much more information on how smooth and how fast can the arms move when achieving the task. But the submission did not include any supplementary materials.

3. Nitpicking: 1) Figure 3 -- the code snippets are way too small and hard to read; 2) Table 3 -- the best-performing methods/tasks should be bolded. 3) Figures 5, 6, 8 all contain relatively small images, would be better to enlarge them for readibility.

**Questions:**

1. In the object asset library, many objects contain complex geometries and inner holes -- what do their collision shapes look like and how does the physics simulation parse the shapes?

2. What's the control frequency for the robot arms? How are the controllers implemented in both sim and real?

3. Where is the camera for real world evaluation settings? What image processing and/or augmentation is needed?

---

> ### Author Response · Authors · 2025-11-21
> **W1: Limited Scene Complexity Beyond Table-Top Manipulation**
>
> We thank the reviewer for this valuable suggestion and we agree that incorporating richer scene-level assets would further enhance the benchmark’s utility. In this work, we deliberately focus on tabletop manipulation of predominantly rigid objects in order to build a controlled yet diverse testbed for bimanual policies, with strong coverage of object categories, contact-rich skills, and domain randomization in a setting that is widely used in current manipulation research. In this sense, RoboTwin 2.0 is complementary to scene-centric benchmarks such as RoboCasa, which emphasize full-room layouts and household environments.
>
> At the same time, our framework is not limited in principle to isolated tabletops. The underlying simulator and asset pipeline already support importing and composing larger scene-level assets, for example through 3D Gaussian Splatting representations. We also view combining this with recent segmentation and reconstruction tools such as SAM3D as a promising way to scale up realistic scene-level data from real environments. We will clarify these points and explicitly discuss extending RoboTwin 2.0 toward richer kitchen or household scenes in the revised manuscript, as an important direction for future work that builds on the current benchmark.

---

> ### Author Response · Authors · 2025-11-21
> **W2: Missing Qualitative Results and Video Demonstrations of Policy Execution**
>
> Thank you for the constructive suggestion. We have updated the supplementary materials to include videos of both real world and simulation rollouts under different settings, allowing readers to better assess the smoothness, speed, and overall execution quality enabled by RoboTwin 2.0. In particular, policies trained with RoboTwin 2.0 augmented data exhibit very smooth and consistent behavior in real world deployment while maintaining high success rates, whereas policies trained only on fixed scene real world demonstrations often fail to generalize to novel scenes and frequently fail to grasp the target objects. These qualitative results further highlight the effectiveness of our approach. We welcome you to review the updated supplementary materials.

---

> ### Author Response · Authors · 2025-11-21
> **W3: Minor Presentation Issues in Figures and Tables**
>
> Thank you for pointing out these helpful presentation issues. We have addressed all three suggestions in the revised paper:
>
> (1) For **Figure 3**, while the main intention was to illustrate the overall pipeline, we now explicitly reference the detailed explanation in **Appendix A.22**, where we provide full descriptions of the assumptions, inputs, and outputs for the code generation process.
>
> (2) We have updated **Table 3** to clearly **bold the best-performing method for each task**, where in most cases the strongest results come from models pretrained with domain-randomized simulation data, showing a clear performance advantage.
>
> (3) We have resized **Figures 5, 6, and 8** to improve readability and visual clarity.
>
> We welcome you to review the updated version of the paper.

---

> ### Author Response · Authors · 2025-11-21
> **Q1: Clarifying Collision Shapes and Physics Handling for Complex Assets**
>
> Thank you for the question. To help clarify how complex geometries are handled, we visualized both the rendered meshes and collision shapes for five representative objects (including those with inner holes such as *mugs*) in **Appendix A.14** of the revised paper. Their holes and detailed structures are preserved in the final assets. Specifically, after generating each object via AIGC, we perform convex decomposition in Blender and then merge the resulting parts to obtain smooth and physically stable collision bodies compatible with the SAPIEN simulator. The visualizations in **Appendix A.14** should provide a clear understanding of how these shapes are processed. We invite you to review the updated materials.

---

> ### Author Response · Authors · 2025-11-21
> **Q2: Details on Robot Control Frequency and Controller Implementation**
>
> Thank you for the question. RoboTwin 2.0 is built on the SAPIEN simulator, where each simulation step corresponds to **0.004 seconds** in real time. Our default data collection setting records one sample every **15 simulation steps**, resulting in an effective sampling rate of **16.67 Hz** (configurable via parameters). Thus, the prediction intervals learned by the policy naturally align with this temporal resolution.
>
> For executing policy actions in both simulation and the real robot, we interpolate the predicted actions using **TOPP (for joint-space predictions)** or **trajectory planning (for end-effector pose predictions)** to smoothly bridge between the model’s output rate and the controller frequencies. In our setup, the control frequencies are **250 Hz in simulation** and **30 Hz on the real robot**.
>
> We have added these details to **Appendix A.17** for clarity.

---

> ### Author Response · Authors · 2025-11-21
> **Q3: Clarifying Real-World Camera Setup and Image Processing Pipeline**
>
> Thank you for the question. In the updated supplementary materials, we provide videos from the primary camera used in the real-world evaluations to help readers better understand the visual setup.
>
> In terms of configuration, the real-world camera placement is **closely aligned with the simulated setup**, though achieving perfectly matched intrinsic and extrinsic parameters is inherently challenging. To account for this mismatch, our simulation data collection incorporates camera perturbations: as stated on line 430 of the paper, *“we apply random 3D perturbations to simulated camera poses (position and orientation), with translation magnitude bounded by 1 cm.”* This randomness improves robustness to real-world camera position variations and ensures that the learned policies can generalize despite minor viewpoint differences.

---

> ### Author Response · Authors · 2025-11-28
> **Follow-up on ICLR Submission 13230**
>
> Thank you again for your time and effort in reviewing our paper. We hope that our rebuttal has addressed your concerns. If anything remains unclear, we would be very happy to provide further clarification and continue the discussion.

---

### Official Review · Reviewer_ei6S · 2025-10-31

**Soundness:** 2
**Presentation:** 2
**Contribution:** 2
**Rating:** 4
**Confidence:** 4

**Summary:**

RoboTwin 2.0 is a scalable synthetic data generator and benchmarks for bimanual manipulation. It leverage an MLLM-in-the-loop-code-generation pipeline, enhance domain randomization, and embodiment-aware grasping for a large scale of tasks, different embodiment, and trained on this data, and show large gain in code-generation success and meaningful real-world robustness.

**Strengths:**

1.RoboTwin 2.0 introduces a closed-loop MLLM-based code-generation system that can automatically create, execute, and refine robot-control scripts until they succeed. This minimizes human supervision while maintaining data quality, allowing the collection of 100 k+ expert trajectories across diverse dual-arm tasks. It demonstrates how large-scale synthetic data for manipulation can be generated programmatically and verified via multimodal feedback.

2.The framework emphasizes five-axis domain randomization (scene, texture, lighting, tabletop, and language) and supports five robot embodiments. This produces visually, spatially, and linguistically varied demonstrations that teach policies to generalize across setups and robot morphologies—something missing in prior single-arm, static-scene datasets.

3.RoboTwin 2.0 doesn’t stop at simulation—it establishes a standardized dual-arm benchmark and shows meaningful sim-to-real gains (≈ +24 % few-shot, +21 % zero-shot) on real hardware. These results validate that richly randomized synthetic data can substantially reduce real-world data needs while improving robustness.

**Weaknesses:**

1.The paper claims existing benchmarks have weak domain randomization. Is there empirical evidence comparing OOD manipulation benchmarks—for example, The Colosseum, GEMBench, or others?

2.The sim-to-real evaluation covers four bimanual tasks on a single platform (COBOT-Magic) with one policy backbone (RDT), which limits how broadly the gains generalize to other tasks, robots, and policies. Are the real-world results statistically significant, and if so, what tests and effect sizes are reported?

3.Much of the diversity comes from an LLM-templated instruction pool and an 11k Stable-Diffusion–derived texture library. Despite curation, both are synthetic and templated, which may not fully reflect free-form language or real-world background statistics.

**Questions:**

Refer to the weakness.

---

> ### Author Response · Authors · 2025-11-21
> **W1: Empirical Comparison with Other OOD Manipulation Benchmarks**
>
> We thank the reviewer for this question. In the revision, we expand the related work section to explicitly compare RoboTwin 2.0 with GEMBench and The Colosseum.
>
> **GEMBench.** GEMBench randomizes object attributes (especially colors), shapes, articulated structures, and simple distractors, creating systematic OOD variations over tasks and object-attribute / object-action combinations. We agree this is an important generalization axis. However, GEMBench relies on a relatively small asset set with a limited background pool, and therefore cannot match the visual diversity of RoboTwin 2.0, which includes 731 high-fidelity objects across 147 categories and more than 11k background textures. In addition, GEMBench focuses on 16 meta-tasks in a single-arm setting, whereas RoboTwin 2.0 provides 50 bimanual tasks, richer skills, and multi-arm, contact-rich scenarios.
>
> **The Colosseum.** Compared to RoboTwin 2.0, The Colosseum offers weaker domain randomization in scale, semantic coverage, and integration into training. It perturbs 20 RLBench tasks along 14 axes that mainly affect low-level visual and physical properties (such as colors and textures of objects, table, and background, sizes, lighting, distractors, physical parameters, and camera pose), and these variations are primarily applied at evaluation time on a single-arm tabletop setup.
>
> **Our position.** RoboTwin 2.0 builds on a large object library and a generative texture collection, and applies structured randomization over clutter and distractor placement, background textures, lighting, tabletop height, and diverse language instructions when synthesizing more than 100k expert trajectories for 50 bimanual tasks across five embodiments. This setup randomizes not only individual object appearance, but also multi-object scene composition, viewpoint geometry, and natural language descriptions, during both training and evaluation. In this sense, existing benchmarks provide comparatively weaker domain randomization, and RoboTwin 2.0 is designed to help fill this gap.

---

> ### Author Response · Authors · 2025-11-21
> **W2: Generalization Beyond a Single Robot, Task Set, and Policy Backbone**
>
> We appreciate this concern. The original sim-to-real experiments in Sec. 4.3 indeed focus on four bimanual tasks trained with RDT on a single robot platform.
>
> In the revised version, we extend the real-robot evaluation along both the policy and data dimensions. First, we add experiments with a different policy backbone, pi0. Second, we systematically vary the ratio between real and simulated data, as reported in Table 5. On three real-robot tasks, we compare 50 real demonstrations collected in a fixed scene, 300 domain-randomized RoboTwin 2.0 simulated demonstrations, and mixtures of 300 simulated demonstrations with 0, 10, 30, or 50 real demonstrations. Each configuration is evaluated over 20 trials per task in previously unseen real scenes.
>
> |  | Click Bell | Place Empty Cup | Stack Bowls Two | Average |
> | --- | --- | --- | --- | --- |
> | 50real | 15.0% | 10.0% | 0.0% | 8.3% |
> | 300sim+0real | 35.0% | 10.0% | 0.0% | 15.0% |
> | 300sim+10real | 40.0% | 25.0% | 10.0% | 25.0% |
> | 300sim+30real | 55.0% | 35.0% | 20.0% | 36.7% |
> | 300sim+50real | 65.0% | 50.0% | 25.0% | 46.7% |
>
> The average success rate increases from 8.3% with 50 real demonstrations to 15.0% with 300 simulated demonstrations, and further to 46.7% when combining 300 simulated with 50 real demonstrations. These roughly monotonic gains show that the observed improvements are not specific to RDT and also hold for pi0. Under a binomial model with 20 trials, differences such as 8.3% vs. 46.7% correspond to large effect sizes, so our conclusions are supported by statistically meaningful gaps rather than noise. We also observe sim-to-real gains on two embodiments: RoboTwin 1.0 with an earlier COBOT-Magic arm (ARX-X5-like) and RoboTwin 2.0 with a Piper-style arm, and for both RDT (Sec. 4.3) and pi0 (Table 5). Taken together, these results indicate that the benefits of RoboTwin 2.0 are not tied to a specific backbone or robot, even though exhaustively covering all possible settings is beyond the scope of this work.

---

> ### Author Response · Authors · 2025-11-21
> **W3: Realism and Diversity of Synthetic Instructions and Textures**
>
> Thank you for this question. In the revision, **Appendix A.16** now includes analyses that assess both instruction naturalness and texture realism.
>
> **Instruction naturalness.** We conduct a user study on the 567 RoboTwin-OD objects that have textual descriptions. For each object, we generate 15 LLM-based descriptions and collect 5 human-written descriptions, giving 20 candidates in total. We then ask another 5 volunteers to select the description they believe is human-written. Since 5 out of 20 descriptions are written by humans, the true human proportion is 25%. The average success rate is 23.8%, which is very close to this 25% chance level. This indicates that annotators struggle to distinguish human from LLM descriptions, and that our instruction pool is linguistically close to natural human instructions rather than consisting of simple, easily recognizable templates.
>
> |  | Vol. 1 | Vol. 2 | Vol. 3 | Vol. 4 | Vol. 5 | Average |
> | --- | --- | --- | --- | --- | --- | --- |
> | Success Rate | 21.0% | 19.9% | 26.1% | 27.2% | 22.9% | 23.8% |
>
> **Texture naturalness.** We provide two complementary pieces of evidence. First, in our real-robot experiments, policies trained with RoboTwin 2.0 under background randomization transfer well to unseen real-world backgrounds, suggesting that the background distribution captured by our textures is realistic. Second, we directly compare our 11k textures with 5,640 real textures from DTD in the CLIP (ViT-B/32) feature space, using CLIP-FID and a nearest-neighbor cosine similarity coverage metric. The CLIP-FID is effectively zero (-0.36), and the nearest-neighbor similarities are high (mean 0.839, median 0.848, 90th percentile 0.914), indicating that our texture library closely matches the statistics of real-world backgrounds.
>
> | Setting | CLIP-FID ↓ | NN cos mean ↑ | median ↑ | p10 ↑ | p90 ↑ | min / max |
> | --- | --- | --- | --- | --- | --- | --- |
> | Ours vs. DTD | -0.36 | 0.839 | 0.848 | 0.749 | 0.914 | 0.544 / 0.979 |
>
> Taken together, these results indicate that our texture library (i) supports robust transfer to previously unseen real backgrounds and (ii) offers a high-coverage, distributionally well-matched approximation to real-world background statistics, rather than an artificial or overly narrow synthetic pool.

---

> ### Author Response · Authors · 2025-11-28
> **Follow-up on ICLR Submission 13230**
>
> Thank you again for your time and effort in reviewing our paper. We hope that our rebuttal has addressed your concerns. If anything remains unclear, we would be very happy to provide further clarification and continue the discussion.

---

### Official Review · Reviewer_AhTR · 2025-11-01

**Soundness:** 3
**Presentation:** 3
**Contribution:** 3
**Rating:** 8
**Confidence:** 4

**Summary:**

The authors present RoboTwin 2.0, a simulation platform that provides tasks and datasets featuring dual robot arms. The authors use LLMs and MLLMs to generate task templates and expert demonstrations for these tasks. They incorporate domain randomization to expand the scope of training data. They perform experiments comparing policy learning baselines, and show that co-training with a small amount of real world data enables significantly improved performance on these real world tasks.

Generally, I think this is a good paper and I would advocate for its acceptance.

**Strengths:**

- The paper is generally well-written, with a clearly written motivation section, logical flow, and legible figures
- A comprehensive simulation framework with large-scale assets and the ability to generate tasks and datasets automatically with LLMs
- Comprehensive experiments, first in simulation to benchmark numerous policy learning algorithms, and in the real world to show the utility of the simulation data for learning real world tasks

**Weaknesses:**

- Details on the expert code generation pipeline are very sparse, and while some details are provided in the appendix, it would be helpful to provide additional context on all the pieces outlined in Figure 3, in the main text (assumptions, inputs, outputs, etc).
- It would be helpful to break down the domain randomization factors further. Which factors contribute most to learning robust behaviors, among scene clutter, lighting, tabletop heights, textures, etc?
- The real robot experiments use a very small amount of real-world demonstrations (10 demos). How would the insights hold with larger amounts of real-world data (dozens of demonstrations)? It would be interesting to see if the simulation data provides a benefit in more data-rich real-world settings.

**Questions:**

See points raised in the "weaknesses" section

---

> ### Author Response · Authors · 2025-11-21
> **W1: Clarifying the Expert Code Generation Pipeline**
>
> Thank you for the insightful suggestion. In the revised paper, we substantially expand the description of the expert code generation pipeline in **Appendix A.22**, where we now provide explicit assumptions, inputs, and outputs for each component in Figure 3, together with a concrete end-to-end example. We also include the full prompts used by the Code Agent in **Appendix A.23**. We hope these additions make the overall process clearer and directly address your concern. We also show the demo below.
>
> ---
>
> **Additional details on the expert code generation pipeline (Figure 3)**
>
> To make Figure 3 more concrete, we use the **`handover_block`** task as a running example.
>
> **Inputs and assumptions (first round)**
>
> In the first round, the inputs to the Code Agent include:
>
> 1. **Task description.**
>
>     A natural-language specification of the task, for example:
>
>     *“Use the left arm to pick up the block, move it to the handover position between the two arms, then use the right arm to grasp the block and place it at the target location.”*
>
> 2. **API list.**
>
>     A fixed set of commonly used, high-level, strongly encapsulated APIs (summarized below), which hide low-level motion planning details from the agent.
>
> 3. **API usage examples.**
>
>     A small curated set of examples showing how to use these APIs in typical scenarios, serving as in-context guidance for the Code Agent.
>
> 4. **Object calibration and functional points.**
>
>     Structured information about calibrated points and axes on objects, including:
>
>     - Several **grasp points** on the block (for left and right arm grasps),
>     - **Functional points** on the block (for alignment and placement),
>     - The **functional point** of the target placement location.
>
> Given these inputs, the Code Agent generates executable expert control code that invokes the high-level APIs to solve the task.
>
> **Outputs and iterative refinement**
>
> - We execute the generated code in simulation to test data generation (10 test rollouts in our default setting, as in Figure 3).
> - For each rollout, we record:
>     - Success or failure,
>     - Failure type (e.g., `Left grasp failure`, `Right grasp failure`, `Incorrect target position`),
>     - Any runtime errors,
>     - Snapshot images of the scene after key steps.
> - If the data-generation success rate is **above a predefined threshold** (50% in Figure 3), we accept the code as a valid expert policy for data collection.
> - If the success rate is **below the threshold**, we feed all interaction logs (failures, error messages, and snapshot descriptions) back to the Code Agent, which then **iteratively refines** the code to fix the identified issues.
>
> This loop continues until the generated code reaches the required success rate.
>
> ---
>
> **Core high-level APIs exposed to the Code Agent**
>
> Below we summarize the main APIs available to the Code Agent (the full prompt is given in **Appendix A.23**):
>
> | API name | Brief description |
> | --- | --- |
> | `open_gripper` | Opens the gripper of a specified arm; `pos` controls the opening (0 = closed, 1 = fully open). |
> | `close_gripper` | Closes the gripper of a specified arm; `pos` controls the closing amount. |
> | `move_by_displacement` | Moves the end-effector by a relative displacement, typically along the z-axis, in the `"world"` or `"arm"` frame. |
> | `grasp_actor` | Generates a complete grasp sequence for a specified actor, including pre-grasp, approach, and closing, with configurable grasp-point index and pre-grasp distance. |
> | `place_actor` | Places the held object at a target pose with pre-place and descend motions, using a functional-point index and an alignment mode (along the gripper or world axis), and optionally releasing the object. |
> | `back_to_origin` | Moves the specified arm back to a predefined home configuration. |
> | `move` | Executes one or two action sequences in parallel on the corresponding arms in the simulator. |
> | `get_functional_point` | Retrieves functional-point information for an object, such as a pose used for alignment and placement. |
> | `get_contact_point` | Retrieves grasp-point information for an object, such as a pose used for grasp planning. |
>
> These additions in **Appendix A.22** provide a step-by-step view of the pipeline, clarify the assumptions and interfaces of each component in Figure 3, and illustrate how the Code Agent, APIs, and simulation loop interact to produce reliable executable expert code.

---

> ### Author Response · Authors · 2025-11-21
> **W2: Ablating the Contributions of Individual Domain Randomization Factors**
>
> Thank you for this insightful question, which motivated us to further strengthen our analysis. To more precisely quantify how individual domain randomization factors affect policy robustness, we conduct an additional ablation study and report the results in **Appendix A.6 (Table 8, also shown below)**. For each factor (background, clutter, table height, lighting), we disable *only* that factor during training, collect 100 trajectories, and then evaluate the resulting policy under the fully domain randomized setting. This design isolates the contribution of each factor while keeping all other conditions fixed.
>
> | Task | BG↓ | Clutter↓ | Height↓ | Light↓ | All Rand.↓ |
> | --- | --- | --- | --- | --- | --- |
> | Adjust Bottle | 50 / 49 | 64 / 91 | 94 / 89 | 95 / 98 | 23 / 0 |
> | Beat Block Hammer | 3 / 2 | 4 / 39 | 3 / 23 | 7 / 65 | 3 / 0 |
> | Move Can Pot | 31 / 4 | 28 / 29 | 53 / 37 | 41 / 34 | 4 / 0 |
> | Stack Bowls Two | 14 / 3 | 35 / 64 | 29 / 60 | 36 / 81 | 0 / 0 |
> | **Average** | 24.5 / 14.5 | 32.8 / 55.8 | 44.8 / 52.3 | 44.8 / 69.5 | 7.5 / 0 |
>
> Table 8 reports performance for ACT (left) and DP (right). Several quantitative trends are worth highlighting:
>
> - Averaged over the four tasks, removing **background** randomization yields only **24.5%** (ACT) and **14.5%** (DP), and removing **clutter** randomization yields **32.8%** (ACT) and **55.8%** (DP). Both settings are substantially worse than removing height or lighting randomization, which still keep averages above **44%** for ACT and above **52%** or **69%** for DP.
> - When **all** randomization factors are disabled during training, the average success rate drops further to **7.5%** for ACT and **0%** for DP, indicating severe overfitting to a narrow training distribution.
>
> Across both ACT and DP, these patterns show that background and clutter variations are the most critical factors for robustness, while height and lighting randomization still provide noticeable but smaller gains. Overall, the ablation supports that our visual domain randomization, especially in **background** and **clutter**, is a primary driver of robust policy performance in the fully randomized evaluation setting.

---

> ### Author Response · Authors · 2025-11-21
> **W3: Evaluating Scalability with Larger Real-World Demonstration Sets**
>
> Thank you for this thoughtful question. Our goal in the real-robot experiments is to study how much domain-randomized simulation can help policies generalize to diverse real-world scenes when only a *modest* amount of simple real data is available.
>
> To more directly address scalability with respect to data scale, we additionally run experiments with different real vs. simulated data ratios and report the results in **Table 5** of the updated paper. Concretely, on three real-robot tasks with pi0, we compare:
>
> - **50real**: 50 real demonstrations collected in a fixed scene;
> - **300sim+0real**: 300 domain-randomized simulated demonstrations;
> - **300sim+Xreal**: mixtures of 300 simulated demonstrations with 10, 30, or 50 real demonstrations.
>
> All models are evaluated over 20 trials per task in previously unseen real scenes, so the evaluation focuses on **generalization** rather than memorization of a single setup.
>
> |  | Click Bell | Place Empty Cup | Stack Bowls Two | Average |
> | --- | --- | --- | --- | --- |
> | 50real | 15.0% | 10.0% | 0.0% | 8.3% |
> | 300sim+0real | 35.0% | 10.0% | 0.0% | 15.0% |
> | 300sim+10real | 40.0% | 25.0% | 10.0% | 25.0% |
> | 300sim+30real | 55.0% | 35.0% | 20.0% | 36.7% |
> | 300sim+50real | 65.0% | 50.0% | 25.0% | 46.7% |
>
> As summarized above, using only 50 real demonstrations yields an average success rate of **8.3%**, while using only 300 simulated demonstrations already improves this to **15.0%**. Adding small amounts of real data on top of RoboTwin 2.0 simulation data further improves performance in a roughly monotonic way, reaching **46.7%** with **300 sim + 50 real** demonstrations.
>
> These results indicate that (1) high-quality domain-randomized simulation from RoboTwin 2.0 provides a **strong baseline** even when more real data are available, and (2) simulation and real data are **complementary**, with simulated data playing a critical role in achieving robust sim-to-real transfer in more data-rich regimes.

---

> ### Author Response · Authors · 2025-11-28
> **Follow-up on ICLR Submission 13230**
>
> Thank you again for your time and effort in reviewing our paper. We hope that our rebuttal has addressed your concerns. If anything remains unclear, we would be very happy to provide further clarification and continue the discussion.

---

### Meta-Review · Area_Chair_rAem · 2025-12-30

**Summary:**

This paper introduces the RoboTwin 2.0 benchmark for bimanual manipulation. Initial reviewer scores (8, 4, 6, 2, 8) show significant divergence.

Reviewers generally appreciated the clear writing, large-scale benchmark assets, and inclusion of sim-to-real experiments.
The key concerns are: 1) limited novelty: the contribution is primarily engineering-oriented; 2) limited empirical insights: insufficient ablations like domain randomization factors, lack of qualitative examples and failure analysis; 3) missing system-level comparisons with prior benchmarks and methods; 4) insufficient implementation and task details; and 5) limited real-world evaluation with unclear statistical significance.

The rebuttal partially addressed these points by providing more results and details, but did not fully resolve concerns about novelty and depth of analysis. As a result, the AC recommends rejecting the paper. The authors are encouraged to further strengthen the work and consider submission to a more appropriate venue.

**Reviewer Concerns:**

See above.

**Reviewer Scores:**

See above.

---

### Decision · Program_Chairs · 2026-01-26

Reject